# Optimizing nitrogen fertilizer amount for best performance and highest economic return of winter wheat under limited irrigation conditions

**Pin Zhang**[1,2,3], **Yi-kang Qi**[1,2,3], **Hong-guang Wang**[1,2,3], **Jian-ning He**[1,2,3], **Rui-qi Li**[1,2,3]*, **Wei-li Liang**[1,3]*

**1** College of Agronomy, Hebei Agricultural University, Baoding, Hebei, China, **2** State Key Laboratory of North China Crop Improvement and Regulation, Baoding, Hebei, China, **3** Key Laboratory of Crop Growth Regulation of Hebei Province, Baoding, Hebei, China

* li-rq69@163.com (LRQ); lwl@hebau.edu.cn (LWL)

**Data Availability Statement:** All relevant data are within the paper and its Supporting information files.

## Abstract

Inappropriate water and fertilizer management can lead to unstable crop yields. Excessive fertilization can potentially cause soil degradation and nitrogen (N) leaching. The aim of this study was to explore the optimal N application rate on two wheat varieties with different nitrogen responding under limited water irrigation at three experimental sites in the Piedmont plain of the Taihang Mountains, China. A two-year field experiment was conducted to explore the effects of five N application rates (N0, N120, N180, N240, and N300) on winter wheat growth, leaf area index, aboveground biomass, grain yield, grain N accumulation, and net return. The results showed that N application rate significantly affected leaf area index, aboveground biomass, grain yield, and harvest index. Variety and variety × N rate interactions had a significant effect on few indicators. Compared with N0, N180 improved leaf area index, aboveground biomass, grain yield, and grain N accumulation. Compared with N240 and N300, N180 increased the harvest index and N harvest index, without significantly reducing grain yield or grain N accumulation, while enhancing a higher N use efficiency. Fertilizers applied in the ranges of 144.7–212.9 and 150.3–247.0 kg ha$^{-1}$ resulted in the highest net return for the KN199 and JM585 varieties, respectively. Our study provides a sound theoretical basis for high-efficiency fertilizer utilization in sustainable winter wheat production in the Piedmont plains of the Taihang Mountains of China.

## Introduction

Wheat (*Triticum aestivum L.*) plays a highly significant role in the nutrition of the Chinese population [1]. Further, the North China Plain (NCP) is one of the primary crop production areas in China, where typical winter wheat is planted in approximately 61% of the total arable land [2]. Winter wheat production in this area plays an important role in ensuring food security and avoiding energy crises in China [3]. However, to continuously maximize crop yield,

**Funding:** National Key Research and Development Program of China (2017YFD0300909).

**Competing interests:** The authors have declared that no competing interests exist.

farmers frequently use nitrogen fertilizers and irrigation excessively. Meanwhile, by and large, groundwater irrigation is the primary irrigation source used across farmlands in the region. Thus, as a result of the excessive exploitation of groundwater, decreasing water availability for agricultural production has gradually restricted the sustainable development of local agriculture [4–6]. Concomitantly, the excessive use of nitrogen fertilizers threatens the ecological safety of the area [7,8] and may increase greenhouse gas emissions [9]. Therefore, increasing winter wheat yield while reducing water and fertilizer inputs, maintaining sustainable crop production, and improving the agroecological environment have all been imperative to achieve the sustainable intensification of agricultural systems in the North China Plain [10].

Sustainable agricultural intensification needs to optimize genotype × environment × management interactions for each target environment [11]. Over the past decades, the development of wheat varieties has contributed greatly to yield increases [12]. Nitrogen absorption by winter wheat is reported significantly correlated with different efficiency varieties [13]. Thirty-nine superior wheat varieties were planted at five N fertilization levels ranging from 0 to 350 kg ha$^{-1}$. The results showed that all interactions between variety and N-rate were highly significant for the total N absorption and N use efficiency [14]. Total N absorption by high-N efficiency cultivars was higher than that of low-N efficiency cultivars [15]. In the NCP, the range of N input is 117–455 kg ha$^{-1}$, but only 35% of the applied N fertilizer is absorbed by winter wheat [16,17], which leads to massive N loss and subsequent negative environmental impacts [18]. Selecting high-efficiency wheat varieties and optimizing agronomic management practices to increase water/nitrogen use efficiency (NUE) is an effective strategy to increase crop yield and reduce associated environmental costs [19]. The most suitable N application rate varies with planting area and wheat variety. Mehrabi et al. [20] showed that there was no significant difference between 150 and 300 kg N ha$^{-1}$ in Shiraz, Iran. The optimal N application rate is 150 kg N ha$^{-1}$. In turn, Xin et al. [12] showed that 150–210 kg N ha$^{-1}$ in winter wheat cultivated in the NCP resulted in higher productivity, higher NUE and lower $N_2O$ emissions.

Although previous studies have optimized fertilizer input to maximize crop yields or net return, few studies have conducted multiple experiments in the Piedmont plains of the Taihang Mountains to investigate the effects of varying N application rates on wheat varieties with differences in factors affecting sustainable agriculture, including N efficiency, wheat yield, N uptake, and net return. Thus, understanding and mastering the N absorption-related in wheat varieties under different N application rates and limited water availability can aid in the development of a reasonable fertilization system to promote sustainable agricultural development, further maintaining crop yield and N absorption, and reducing soil erosion hazards. In this study, a two-year field experiment was conducted on winter wheat at three locations to determine the effects of the use of varieties and N application rates on wheat growth, grain yield, grain N accumulation, and net return. Additionally, the appropriate winter wheat varieties and N fertilization ranges to simultaneously optimize the grain yield, grain N accumulation, and net return were determined. The results of this study provide a technical reference for the production of high winter-wheat yields while promoting the synergistic utilization of varieties and nutrients in similar plains around the world.

## Materials and methods

### Experimental site description

A field experiment was conducted in three representative locations in the Piedmont plain during the 2018–2019 and 2019–2020 winter wheat growing seasons in Gaocheng Liujiazhuang Village (37˚96' N, 114˚89' E), Xinle Zhongtong Village (38˚40' N, 114˚71' E), and Zhaoxian Northern Agricultural Park (38˚40' N, 114˚71' E). The study site in Gaocheng has a temperate,

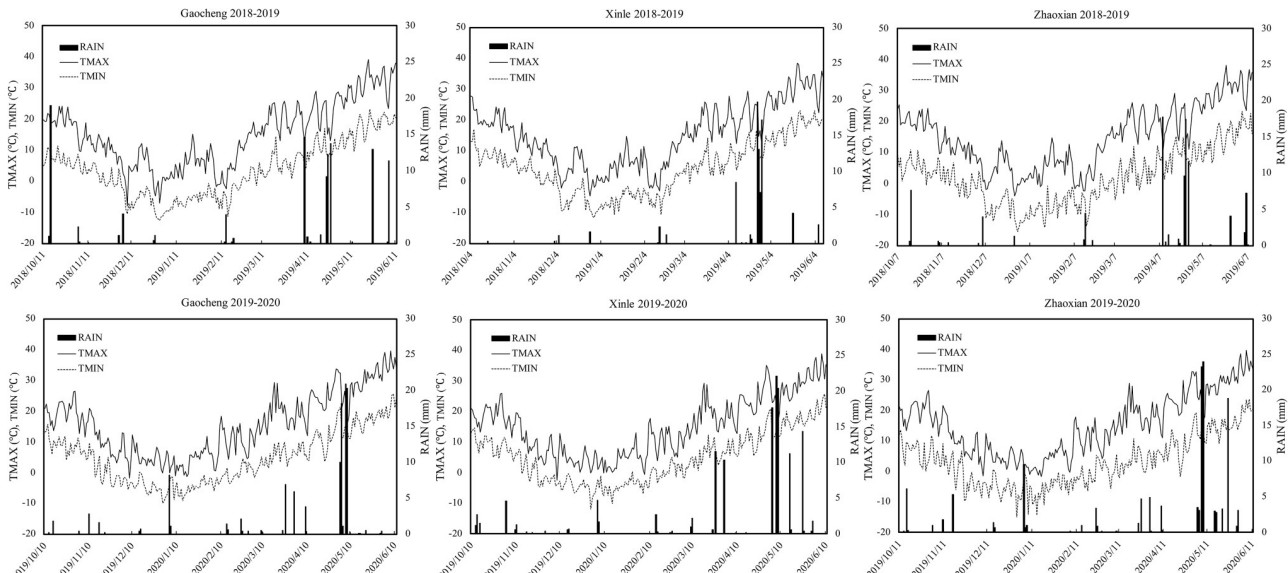

**Fig 1. Daily rainfall (RAIN), maximum temperature (TMAX) and minimum temperature (TMIN) in three experimental sites during 2018–2020 seasons.**

semi-humid, and continental monsoon climate, with an annual average temperature of 12.5˚C and an average annual precipitation of 494.0 mm. On the other hand, the study site in Xinle is located at the eastern foot of the Taihang Mountains, on the sloped plain in front of the mountains. The site has a warm temperate, semi-humid monsoon continental climate, with an average temperature of 12.3˚C and an average annual precipitation of 428.9 mm. In turn, the study site in Zhaoxian is located on the Piedmont alluvial plain in the middle section of the eastern foothills of the Taihang Mountains. This site has a warm temperate, semi-humid monsoon continental climate with an annual average temperature of 13˚C and an average annual precipitation of 502.5 mm. The total amounts of precipitation during the wheat growing season in 2018–2019 and 2019–2020 were 94.4 mm and 135.2 mm at Zhaoxian, 111.4 mm and 112.0 mm at Gaocheng, and 81.3 mm and 139.4 mm at Xinle, respectively. Monthly total precipitations and mean temperatures are shown in Fig 1. The main characteristics of the 0–20 cm and 0–40 cm soil layers in the experimental region at the beginning of the experiment are shown in Table 1.

## Experimental design

The experiment adopted a two-factor split plot design at three experimental locations. The main plots were arranged for varieties KN199 (low nitrogen tolerant and high yield) and JM585 (nitrogen sensitive) previously studied [21]. The subplots were arranged for N application with five N fertilizer rates, namely, 0, 120, 180, 240, and 300 kg N ha$^{-1}$ (referred to as N0, N120, N180, N240, and N300, respectively). Main plots were 50 m long ×10 m wide and subplots were 9 m long ×5 m wide in size, with a 1.0-m buffer zone between plots to minimize the effects of adjacent plots. In sum, there were 30 treatments with three replicates each. Both wheat varieties were sown at a rate of 180 kg seed ha$^{-1}$. At the third-leaf growth stage, seedlings are thinned to attain a plant population density of 3.3 million basic seedlings per hectare. In the 2018–2019 winter growing season, plots in Gaocheng, Xinle, and Zhaoxian were planted

**Table 1. The status of top soil before seeding in 2018 and 2019.**

| Growing season | Soil layer (cm) | Experiment Location | Soil bulk density (g cm$^{-3}$) | Organic matter (g kg$^{-1}$) | Available N (mg kg$^{-1}$) | Available P (mg kg$^{-1}$) | Available K (mg kg$^{-1}$) |
|---|---|---|---|---|---|---|---|
| 2018–2019 | 0–20 | Gaocheng | 1.64 | 21.30 | 146.42 | 22.50 | 169.60 |
| | | Xinle | 1.78 | 19.58 | 118.42 | 10.40 | 108.20 |
| | | Zhaoxian | 1.66 | 20.42 | 135.92 | 22.10 | 152.30 |
| | 20–40 | Gaocheng | 1.67 | 5.33 | 60.00 | 5.20 | 146.80 |
| | | Xinle | 1.76 | 4.89 | 56.58 | 1.21 | 91.70 |
| | | Zhaoxian | 1.73 | 5.02 | 53.08 | 2.70 | 111.20 |
| 2019–2020 | 0–20 | Gaocheng | 1.65 | 20.73 | 140.75 | 22.59 | 162.92 |
| | | Xinle | 1.72 | 17.58 | 118.73 | 10.91 | 112.95 |
| | | Zhaoxian | 1.68 | 18.98 | 133.43 | 21.53 | 149.99 |
| | 20–40 | Gaocheng | 1.69 | 5.05 | 56.70 | 5.33 | 147.76 |
| | | Xinle | 1.75 | 4.44 | 53.57 | 1.42 | 94.73 |
| | | Zhaoxian | 1.74 | 4.87 | 51.58 | 2.61 | 114.26 |

on October 11, 4, and 7, respectively, and the crops were harvested on June 11, 4, and 7, respectively. Meanwhile, in the 2019–2020 growing season, plots in Gaocheng, Xinle, and Zhaoxian were planted on October 10, 10 and 11, respectively, and the crops were harvested on June 15, 10 and 11, respectively. This experiment was carried out under limited water conditions: during the whole growing season, only 60 mm water were applied at the jointing stage. N fertilizer was spread before irrigation at jointing stage, a water meter was connected at the water outlet of the pump, then the water pipe was connected to the plots in order to strictly control water quantity at 60 mm.

Nitrogen, phosphorus and potassium fertilizers were applied using urea (46.7% N), calcium superphosphate (16% $P_2O_5$), and potassium sulfate (60% $K_2O$), respectively. The base fertilization before seeding was applied by spreading the fertilizer on the surface and mixing it into the soil with a rotary cultivator. P and K fertilizers were applied at 120 kg ha$^{-1}$ and 150 kg ha$^{-1}$, respectively, as base fertilizers; in turn, N fertilizer was split in two portions applied as base fertilizer (50%) and topdressing (50%) at the jointing stage. Herbicides (spray 15% Tribenuron 10 g hm$^{-2}$) were applied before sowing and pesticides (mix 10% imidacloprid with 50 kg of water and spray) were applied at flowering period, so that the plots were kept free of weeds, insects, and diseases during the growing seasons.

## Measurements and calculations

**Leaf area index (LAI).** At anthesis, the leaf area of 20 representative plants from each plot was measured using a LI-3000C portable leaf area meter (LI-COR, Lincoln Nebraska, USA).

**Aboveground biomass.** At maturity, 20 representative plants were selected from each plot. Each plant was divided into grain, stem and sheath, leaves, and the spike axis and glumes, and oven-dried at 105°C for 30 min and subsequently dried at 75°C to constant weight. The plant mass was determined using an electronic balance. The total accumulation of dry matter of each organ per hectare was calculated by sampling the number of plants and the basic seedling number per hectare.

**Grain yield and harvest index.** At grain maturity, an area of 2 m$^2$ of undisturbed wheat was randomly selected to measure yield. Grains were air-dried and weighed after harvest, samples were taken and oven-dried for measuring moisture content. Yields were normalized at

13% moisture content.

$$\text{Harvest index (HI)} = \frac{\text{Grain yield (kg ha}^{-1})}{\text{Aboveground biomass (kg ha}^{-1})} \tag{1}$$

**N accumulation, N harvest index and N use efficiency.** The various organs of the wheat plants whose dry matter was measured at maturity were ground to determine N accumulation. Dried samples were ground, extracted with $H_2SO_4$-$H_2O_2$, and analyzed for total N content (%) using a continuous-flow auto-analyzer (AutoAnalyzer 3; Bran + Luebbe, Noderstedt, Germany) [22].

Total N accumulation, N harvest index (NHI) and NUE were calculated according to the method described by Zhang et al. [23].

$$\text{Total N accumulation (kg ha}^{-1}) = \text{N content (\%)} \times \text{aboveground biomass (kg ha}^{-1}) \tag{2}$$

$$\text{NHI} = \frac{\text{Grain N accumulation (kg ha}^{-1})}{\text{Total N accumulation (kg ha}^{-1})} \tag{3}$$

$$\text{NUE} = \frac{\text{Grain yield (kg ha}^{-1})}{\text{Total N accumulation(kg ha}^{-1})} \tag{4}$$

**Net return.** Net return was calculated as follows [24]:

$$\text{Nr} = \text{Gp} - \text{Ic} - \text{Fc} - \text{O} \tag{5}$$

where, Nr is the net return (CNY ha$^{-1}$), Gp is the gross profit (CNY ha$^{-1}$), Ic is the irrigation cost (CNY ha$^{-1}$), Fc is the fertilizer cost (CNY ha$^{-1}$), and O represents other costs (CNY ha$^{-1}$).

## Data analysis

Analysis of variance (ANOVA) was performed using IBM SPSS Statistics for Windows, version 25 (IBM Corp., Armonk, NY, USA). Variety and fertilization rate were used as the primary effects and two-way interactions were included as well. All treatment means were compared for significant differences using Duncan's multiple range test at a significance level of $P = 0.05$ [25].

## Results

### Leaf area index, aboveground biomass, grain yield, and harvest index

Location, variety and N rate interaction significantly affected LAI in 2018–2020. The interaction of L×V significantly affected the LAI in 2018–2020, and L×V only affected the LAI in 2019–2020 (Table 2). As shown in Fig 2, LAI recorded for KN199 under the N180 treatment increased significantly by 24.11%, 63.10%, and 45.79% at Gaocheng, Xinle, and Zhaoxian, respectively, compared with the N0'treatment in 2018–2019 growing season. In 2018–2019, LAI for JM585 under the N180 treatment increased significantly by 15.08%, 82.89%, and 42.11% at Gaocheng, Xinle, and Zhaoxian, respectively, compared with the N0'treatment. Moreover, in 2019–2020, at Xinle and Zhaoxian experimental locations, the LAI of JM585 increased significantly by 4.34% and 11.95%, respectively when the N input increased from N180 to N240.

**Table 2. Significance level (*P* values) of the effects of location, variety and N rate on leaf area index, aboveground biomass, grain yield, HI, total N accumulation, grain N accumulation, NHI and NUE of winter wheat.**

| Growing season | Treatment | Leaf area index | Aboveground biomass | Grain yield | HI | Total N accumulation | Grain N accumulation | NHI | NUE |
|---|---|---|---|---|---|---|---|---|---|
| 2018–2019 | Location (L) | <0.001 | <0.001 | <0.001 | <0.001 | <0.001 | <0.001 | <0.001 | <0.001 |
| | Variety (V) | <0.001 | <0.001 | 0.021 | <0.001 | 0.547 | <0.001 | <0.001 | 0.017 |
| | N rate (N) | <0.001 | <0.001 | <0.001 | <0.001 | <0.001 | <0.001 | <0.001 | <0.001 |
| | L×V | <0.001 | <0.001 | 0.536 | <0.001 | <0.001 | <0.001 | <0.001 | 0.005 |
| | L×N | 0.983 | <0.001 | 0.090 | <0.001 | <0.001 | <0.001 | <0.001 | <0.001 |
| | V×N | 0.365 | 0.001 | 0.845 | 0.196 | <0.001 | <0.001 | 0.099 | <0.001 |
| | L×V×N | 0.433 | 0.003 | 0.999 | 0.021 | 0.128 | 0.178 | 0.013 | <0.001 |
| 2019–2020 | Location (L) | <0.001 | <0.001 | <0.001 | 0.170 | <0.001 | <0.001 | <0.001 | <0.001 |
| | Variety (V) | <0.001 | 0.004 | <0.001 | <0.001 | <0.001 | <0.001 | <0.001 | <0.001 |
| | N rate (N) | <0.001 | <0.001 | <0.001 | <0.001 | <0.001 | <0.001 | <0.001 | <0.001 |
| | L×V | <0.001 | 0.005 | <0.001 | <0.001 | 0.557 | 0.005 | <0.001 | <0.001 |
| | L×N | <0.001 | 0.002 | 0.042 | 0.093 | <0.001 | <0.001 | <0.001 | <0.001 |
| | V×N | 0.820 | 0.017 | 0.426 | 0.170 | 0.132 | 0.302 | 0.832 | 0.018 |
| | L×V×N | 0.199 | 0.003 | 0.101 | 0.849 | 0.003 | 0.003 | 0.021 | 0.471 |
| 2018–2020 | Growing season (S) | <0.001 | <0.001 | <0.001 | <0.001 | <0.001 | <0.001 | <0.001 | <0.001 |
| | Location (L) | <0.001 | <0.001 | <0.001 | <0.001 | <0.001 | <0.001 | <0.001 | <0.001 |
| | Variety (V) | <0.001 | <0.001 | <0.001 | <0.001 | <0.001 | <0.001 | <0.001 | <0.001 |
| | N rate (N) | <0.001 | <0.001 | <0.001 | <0.001 | <0.001 | <0.001 | <0.001 | <0.001 |
| | S×L | <0.001 | <0.001 | <0.001 | 0.001 | <0.001 | <0.001 | <0.001 | <0.001 |
| | S×V | <0.001 | <0.001 | <0.001 | <0.001 | <0.001 | <0.001 | <0.001 | <0.001 |
| | S×N | <0.001 | <0.001 | <0.001 | 0.005 | <0.001 | 0.216 | <0.001 | 0.019 |
| | L×V | <0.001 | <0.001 | <0.001 | <0.001 | 0.001 | <0.001 | <0.001 | <0.001 |
| | L×N | <0.001 | <0.001 | 0.002 | 0.004 | <0.001 | <0.001 | <0.001 | <0.001 |
| | V×N | 0.378 | <0.001 | 0.313 | 0.023 | <0.001 | <0.001 | 0.911 | 0.036 |
| | S×L×V | <0.001 | 0.143 | <0.001 | <0.001 | 0.109 | 0.264 | 0.013 | <0.001 |
| | S×L×N | <0.001 | <0.001 | 0.672 | 0.007 | <0.001 | <0.001 | <0.001 | 0.197 |
| | S×V×N | 0.686 | 0.856 | 0.961 | 0.871 | 0.033 | 0.078 | 0.110 | <0.001 |
| | L×V×N | 0.466 | 0.053 | 0.399 | 0.352 | 0.005 | 0.009 | 0.017 | <0.001 |
| | S×L×V×N | 0.265 | <0.001 | 0.522 | 0.188 | 0.020 | 0.013 | 0.013 | <0.001 |

Note: HI: Harvest index; NHI: Nitrogen Harvest index; NUE: Nitrogen use efficiency.

Location, variety, and N rate significantly affected aboveground biomass in the two growing seasons considered herein. The interaction of L×V, L×N and V×N significantly affected the aboveground biomass in 2018–2020 (Table 2). As shown in Fig 3, the aboveground biomass of KN199 under the N180 treatment increased significantly by 23.12%, 25.60%, and 17.66% at Gaocheng, Xinle, and Zhaoxian, respectively, compared with the N0'treatment in 2018–2019 growing season. In the 2018–2019 growing season, the aboveground biomass of JM585 increased significantly by 24.69%, 26.56%, and 20.37% at Gaocheng, Xinle, and Zhaoxian, respectively, under the N180 treatment, compared to the N0'treatment. Furthermore, when N input increased to N240 and N300, aboveground biomass significantly increased in Xinle and Zhaoxian experimental locations; However, grain yield did not increase significantly, while dry matter accumulated in vegetative organs was the portion of the plant body whose dry matter increased more. In the 2019–2020 growing season, grain yield and dry matter for the two winter wheat varieties were consistent with those observed in the previous season.

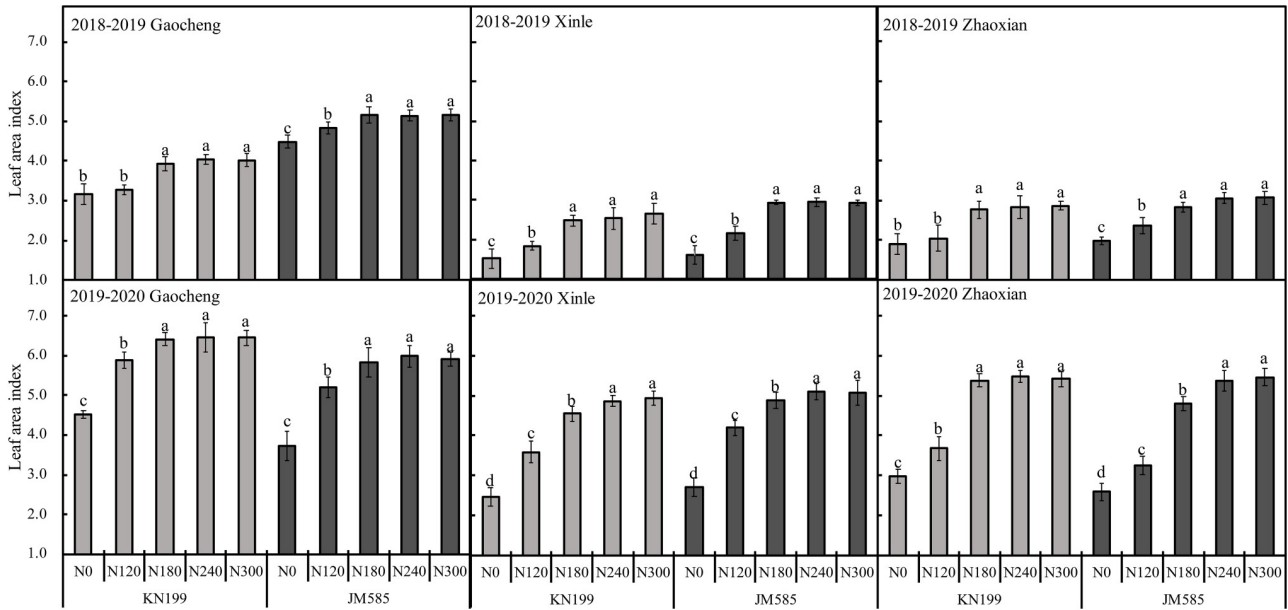

**Fig 2. Leaf area index at anthesis of different winter wheat varieties under different nitrogen fertilization rates in three experimental sites during 2018–2020 seasons.** Error bars represent one standard deviation from the mean. Letters above the bars are comparison results of leaf area index between different nitrogen treatments of the same variety. There is no significant difference between treatments with same letters.

Location, variety, and N rate significantly affected grain yield in the two growing seasons considered herein. The interaction of L×V significantly affected the grain yield in 2019–2020 (Table 2). Thus, as shown in Fig 4, grain yield of KN199 increased significantly by 14.70%, 14.07%, and 9.16% at Gaocheng, Xinle, and Zhaoxian, respectively, under N180 compared to the N0'treatment in 2018–2019 growing season. Similarly, in 2018–2019, grain yield of JM585

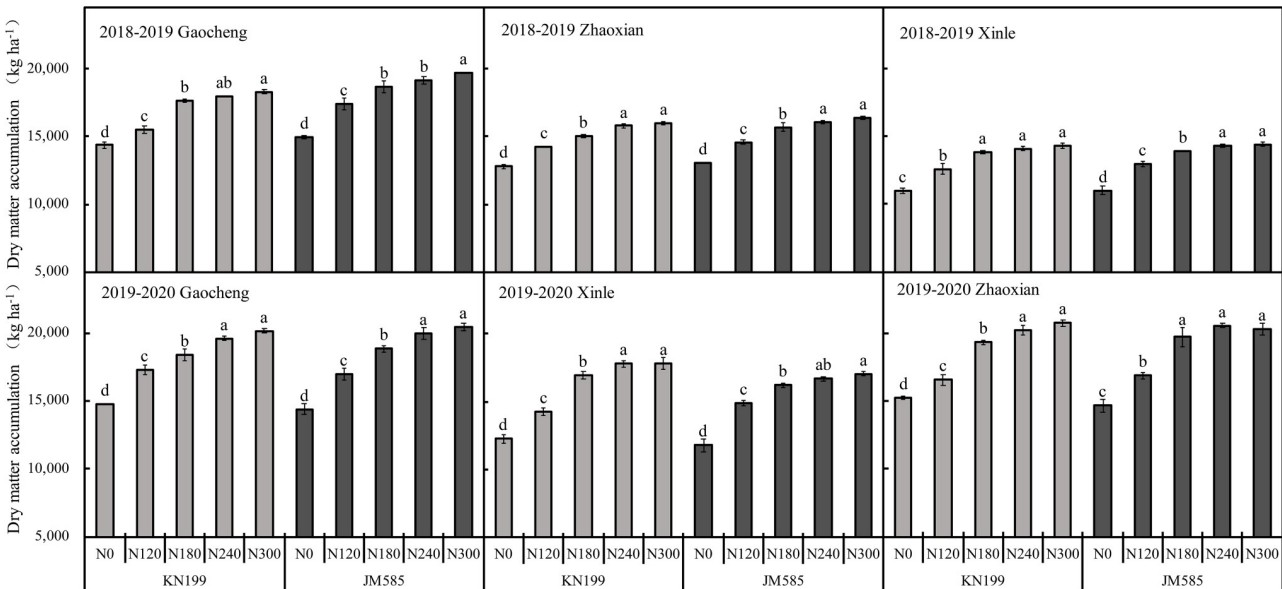

**Fig 3. Aboveground biomass at maturity of different winter wheat varieties under different nitrogen fertilization rates in three experimental sites during 2018–2020 seasons.** Error bars represent one standard deviation from the mean. Letters above the bars are comparison results of leaf area index between different nitrogen treatments of the same variety. There is no significant difference between treatments with same letters.

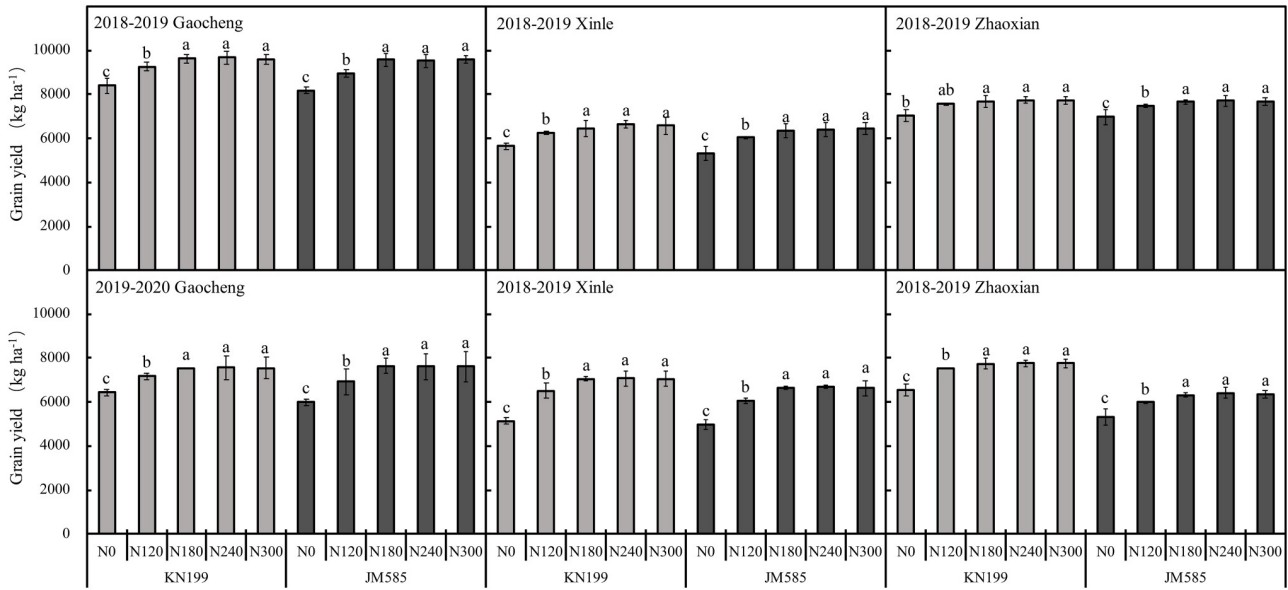

**Fig 4. Grain yield of different winter wheat varieties under different nitrogen fertilization rates in three experimental sites during 2018–2020 seasons.** Error bars represent one standard deviation from the mean. Letters above the bars are comparison results of leaf area index between different nitrogen treatments of the same variety. There is no significant difference between treatments with same letters.

increased significantly by 16.99%, 18.96%, and 9.72% at Gaocheng, Xinle, and Zhaoxian, respectively, under N180 compared to the N0'treatment. Furthermore, in 2019–2020, the grain yield variation observed for the two winter wheat varieties evaluated was consistent with that observed in the 2018–2019 season. On the other hand, in 2019–2020 at Gaocheng, the grain yield of KN199 and JM585 under the N0–N300 treatments significantly lower than 2018–2019. No significant differences in grain yield were detected among N180, N240, and N300 in either growing season.

Variety and N rate significantly affected HI in 2018–2020 while location significantly affected HI in 2018–2019. The interaction of L×V significantly affected the HI in 2018–2020, and L×N only affected the HI in 2019–2020 (Table 2). As shown in Fig 5, HI was highest under N0 and decreased as the N application rate increased. The range of HI values observed in 2018–2019 for KN199 under N0–N300 at Gaocheng, Xinle, and Zhaoxian, was 0.48–0.50, 0.47–0.50, and 0.47–0.51, respectively. Similarly, in 2018–2019, the range HI values observed that same year for JM585 HI was 0.45–0.46, 0.45–0.47, and 0.45–0.50, respectively. In 2019–2020, HI values for the two varieties of winter wheat were consistent with those observed in 2018–2019. HI values for KN199 were higher than those for JM585 across N application rates.

## Winter wheat nitrogen accumulation, N harvest index, and NUE

Location and N rate significantly affected total N accumulation in 2018–2020 while variety significantly affected total N accumulation in 2019–2020. The interaction of L×V significantly affected the total N accumulation in 2018–2020, and L×N, V×N only affected the total N accumulation in 2018–2019. Overall, location, variety, and N rate significantly affected grain N accumulation in 2018–2020. The interaction of L×V and L×N significantly affected the grain N accumulation in 2018–2020, and V×N only affected the total N accumulation in 2018–2019. Location, variety, and N rate significantly affected NHI in 2019–2020. The interaction of L×V and L×N significantly affected the NHI in 2018–2020. Location, variety and N rate in

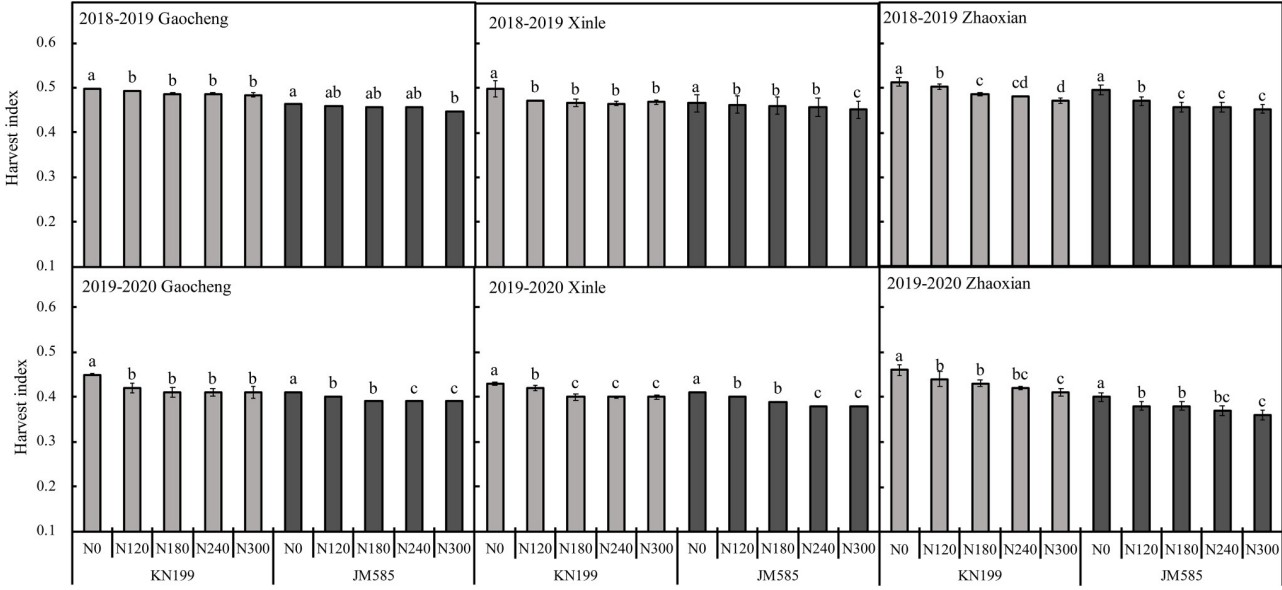

**Fig 5. Harvest index of different winter wheat varieties under different nitrogen fertilization rates in three experimental sites during 2018–2020 seasons.** Error bars represent one standard deviation from the mean. Letters above the bars are comparison results of leaf area index between different nitrogen treatments of the same variety. There is no significant difference between treatments with same letters.

2018–2020. The interaction of L×V, L×N and V×N significantly affected NUE in 2018–2020 (Table 2).

As shown in Table 3, total N accumulation in KN199 increased significantly by 28.09%, 28.00%, and 37.35% at Gaocheng, Xinle, and Zhaoxian, respectively, under N180 compared to the N0'treatment in 2018–2019 growing season. Similarly, in 2018–2019, total N accumulation in JM585 increased significantly by 40.16%, 34.83%, and 45.68% at Gaocheng, Xinle, and Zhaoxian, respectively, under N180 compared to the N0'treatment. Furthermore, when N input increased to N240, total N accumulation significantly increased in the three experimental locations. When N input increased to N300, the total N accumulation did not increase significantly. Grain N accumulation in KN199 increased significantly by 23.97%, 23.76%, and 28.41% at Gaocheng, Xinle, and Zhaoxian, respectively, under N180 compared to the N0'treatment in 2018–2019 growing season. Similarly, in 2018–2019, grain N accumulation in JM585 increased significantly by 40.16%, 34.83%, and 45.68% at Gaocheng, Xinle, and Zhaoxian, respectively, under N180 compared to the N0'treatment. NHI for KN199 was 0.86–0.83, 0.89–0.85, and 0.89–0.81 at Gaocheng, Xinle, and Zhaoxian, respectively, while NHI for JM585 was 0.85–0.81, 0.86–0.83, and 0.86–0.79 at Gaocheng, Xinle, and Zhaoxian, respectively. Furthermore, in 2019–2020, total N accumulation, grain N accumulation, and NHI and NUE variation for the two winter wheat varieties tested were consistent with the corresponding values obtained in the previous season. Total N accumulation and grain N accumulation increased with increasing fertilization rates, whereas NHI and NUE decreased with increasing fertilization rates.

## Economic return

As shown in Table 4, In 2018–2020, water cost and cost of cultivation for two winter wheat varieties under five N treatment being same at three experiment location. Fertilizer cost increased as fertilizer rate increased in both winter wheat varieties. In 2018–2020, fertilizer

**Table 3. Nitrogen accumulation at maturity of different winter wheat varieties under different nitrogen fertilization rates in three experimental sites during 2018–2020 seasons.**

| 2018–2019 | | | | | | | 2019–2020 | | | |
|---|---|---|---|---|---|---|---|---|---|---|
| Treatment | | | Total nitrogen accumulation (kg ha$^{-1}$) | Grain nitrogen accumulation (kg ha$^{-1}$) | N harvest index | NUE | Total nitrogen accumulation (kg ha$^{-1}$) | Grain nitrogen accumulation (kg ha$^{-1}$) | N harvest index | NUE |
| Location | Variety | N rate | | | | | | | | |
| Gaocheng | KN199 | N0 | 208.17d | 178.19d | 0.86a | 34.36a | 201.26d | 171.25c | 0.85a | 33.18a |
| | | N120 | 231.40c | 195.04c | 0.84b | 33.00b | 242.03c | 190.82b | 0.79b | 29.81b |
| | | N180 | 266.65b | 220.90b | 0.83c | 32.91b | 258.91b | 198.13b | 0.77c | 28.91bc |
| | | N240 | 292.54a | 243.09a | 0.83c | 30.44c | 283.22a | 213.29a | 0.75d | 28.14c |
| | | N300 | 296.58a | 244.71a | 0.83c | 30.40c | 291.60a | 217.45a | 0.75d | 28.11c |
| | JM585 | N0 | 201.29d | 170.64c | 0.85a | 34.41a | 181.94d | 148.78d | 0.82a | 32.66a |
| | | N120 | 247.49c | 204.98b | 0.83b | 31.95b | 222.93c | 174.15c | 0.78b | 30.35b |
| | | N180 | 282.13b | 232.52a | 0.82b | 30.24c | 258.54b | 192.22b | 0.74c | 28.26c |
| | | N240 | 295.50a | 240.03a | 0.81c | 29.57c | 281.94a | 207.79a | 0.74c | 27.69c |
| | | N300 | 300.91a | 243.06a | 0.81c | 29.20c | 287.73a | 210.27a | 0.73c | 27.48c |
| Xinle | KN199 | N0 | 143.98d | 128.34d | 0.89a | 38.07a | 148.78d | 124.87d | 0.84a | 35.87a |
| | | N120 | 169.38c | 148.08c | 0.87b | 35.04b | 186.70c | 154.67c | 0.83b | 32.17b |
| | | N180 | 184.30b | 158.84b | 0.86c | 35.05b | 208.74b | 167.43b | 0.80c | 32.61b |
| | | N240 | 203.01a | 173.65a | 0.86c | 32.81c | 245.31a | 195.19a | 0.80c | 28.84c |
| | | N300 | 208.88a | 178.14a | 0.85c | 32.35c | 249.41a | 196.26a | 0.79d | 28.46c |
| | JM585 | N0 | 134.23d | 116.06d | 0.86a | 38.26a | 138.10d | 113.82d | 0.82a | 34.85a |
| | | N120 | 169.12c | 145.17c | 0.86a | 35.54b | 183.43c | 146.65c | 0.80b | 32.37b |
| | | N180 | 180.98b | 154.68b | 0.85b | 35.40b | 209.17b | 163.54b | 0.78c | 30.56c |
| | | N240 | 197.95a | 167.60a | 0.85b | 33.02c | 223.92a | 170.77a | 0.76d | 28.43d |
| | | N300 | 201.67a | 170.05a | 0.84c | 32.03c | 226.02a | 171.64a | 0.76d | 28.52d |
| Zhaoxian | KN199 | N0 | 164.44d | 145.98c | 0.89a | 39.89a | 167.47d | 142.30c | 0.85a | 41.99a |
| | | N120 | 184.18c | 160.56b | 0.87b | 38.87b | 205.01c | 172.88b | 0.84a | 35.27b |
| | | N180 | 225.87b | 187.46a | 0.83c | 32.38c | 259.01b | 208.55a | 0.81b | 32.39c |
| | | N240 | 241.76ab | 196.98a | 0.81d | 31.35d | 276.01a | 215.54a | 0.78c | 30.98c |
| | | N300 | 243.60a | 198.11a | 0.81d | 30.90d | 279.44a | 216.17a | 0.77c | 30.75c |
| | JM585 | N0 | 152.77d | 131.23c | 0.86a | 42.29a | 155.75d | 126.84c | 0.81a | 37.78a |
| | | N120 | 200.17c | 167.37b | 0.84b | 34.26b | 196.83c | 154.54b | 0.78b | 32.47b |
| | | N180 | 222.56b | 178.18a | 0.80c | 32.21c | 246.35b | 184.17a | 0.75c | 29.01c |
| | | N240 | 228.46ab | 181.36a | 0.79d | 32.13c | 258.87ab | 188.80a | 0.73d | 28.02cd |
| | | N300 | 237.42a | 187.44a | 0.79d | 31.24c | 269.72a | 194.66a | 0.72d | 27.22d |

Note: Different letters following data of same trait and variety indicate significant differences between nitrogen treatments ($P < 0.05$).

cost for two winter wheat varieties under the N180 treatment being higher by 13%, 50% than N0, N120, being lower by 10%, 22% than N240, N300 treatment at three experiment location. Gross profit obtained from the two tested winter wheat varieties increased with N application rate across locations and over the two growing seasons. However, no significant further increases in gross profit were observed when N input exceed to 180 kg ha$^{-1}$. Net return first increased and then decreased as fertilizer rate increased in both winter wheat varieties. In 2018–2019, net return obtained from KN199 was significantly the highest under the N180 treatment being higher by 1.49%–13.76%, 1.20%–16.21% than other treatment at Gaocheng and Xinle, under the N120 treatment being higher by 0.53%–6.14% than other treatment at Zhaoxian. Net return obtained from JM585 was significantly the highest under the N180 treatment being higher by 3.41%–19.25%, 3.36%–27.14%, 0.56%–6.85% than other treatment at

**Table 4. Economic return (CNY ha⁻¹) of different winter wheat varieties under different nitrogen fertilization rates in three experimental sites during 2018–2020 seasons.**

| Treatment | | | 2018–2019 | | | | | 2019–2020 | | | | |
|---|---|---|---|---|---|---|---|---|---|---|---|---|
| | | | Water cost | Fertilizer cost | Cost of cultivation | Gross profit | Net return | Water cost | Fertilizer cost | Cost of cultivation | Gross profit | Net return |
| Location | Variety | N rate | | | | | | | | | | |
| Gaocheng | KN199 | N0 | 1060a | 1725e | 3600a | 16778b | 10393b | 1060a | 1725e | 3600a | 12889b | 6504b |
| | | N120 | 1060a | 2297d | 3600a | 18519a | 11562a | 1060a | 2297d | 3600a | 14333a | 7376ab |
| | | N180 | 1060a | 2585c | 3600a | 19244a | 11999a | 1060a | 2585c | 3600a | 15081a | 7836a |
| | | N240 | 1060a | 2873b | 3600a | 19356a | 11823a | 1060a | 2873b | 3600a | 15105a | 7572ab |
| | | N300 | 1060a | 3159a | 3600a | 19174a | 11355a | 1060a | 3159a | 3600a | 15096a | 7277ab |
| | JM585 | N0 | 1060a | 1725e | 3600a | 16348c | 9963c | 1060a | 1725e | 3600a | 11185c | 4800c |
| | | N120 | 1060a | 2297d | 3600a | 17889b | 10932b | 1060a | 2297d | 3600a | 13230b | 6273bc |
| | | N180 | 1060a | 2585c | 3600a | 19126a | 11881a | 1060a | 2585c | 3600a | 15259a | 8014a |
| | | N240 | 1060a | 2873b | 3600a | 19022a | 11489ab | 1060a | 2873b | 3600a | 15214a | 7681ab |
| | | N300 | 1060a | 3159a | 3600a | 19154a | 11335ab | 1060a | 3159a | 3600a | 15207a | 7388ab |
| Xinle | KN199 | N0 | 1060a | 1725e | 3600a | 11319b | 4934a | 1060a | 1725e | 3600a | 10257c | 3872b |
| | | N120 | 1060a | 2297d | 3600a | 12533a | 5576a | 1060a | 2297d | 3600a | 13029b | 6072a |
| | | N180 | 1060a | 2585c | 3600a | 12911a | 5666a | 1060a | 2585c | 3600a | 14111a | 6866a |
| | | N240 | 1060a | 2873b | 3600a | 13267a | 5734a | 1060a | 2873b | 3600a | 14148a | 6615a |
| | | N300 | 1060a | 3159a | 3600a | 13148a | 5329a | 1060a | 3159a | 3600a | 14096a | 6277a |
| | JM585 | N0 | 1060a | 1725e | 3600a | 10667b | 4282a | 1060a | 1725e | 3600a | 9968c | 3583c |
| | | N120 | 1060a | 2297d | 3600a | 12067a | 5110a | 1060a | 2297d | 3600a | 12111b | 5154b |
| | | N180 | 1060a | 2585c | 3600a | 12689a | 5444a | 1060a | 2585c | 3600a | 13278a | 6033a |
| | | N240 | 1060a | 2873b | 3600a | 12800a | 5267a | 1060a | 2873b | 3600a | 13389a | 5856ab |
| | | N300 | 1060a | 3159a | 3600a | 12884a | 5065a | 1060a | 3159a | 3600a | 13252a | 5433ab |
| Zhaoxian | KN199 | N0 | 1060a | 1725e | 3600a | 14067b | 7682a | 1060a | 1725e | 3600a | 13074b | 6689b |
| | | N120 | 1060a | 2297d | 3600a | 15111a | 8154a | 1060a | 2297d | 3600a | 15044a | 8087a |
| | | N180 | 1060a | 2585c | 3600a | 15356a | 8111a | 1060a | 2585c | 3600a | 15481a | 8236a |
| | | N240 | 1060a | 2873b | 3600a | 15452a | 7919a | 1060a | 2873b | 3600a | 15513a | 7980a |
| | | N300 | 1060a | 3159a | 3600a | 15450a | 7631a | 1060a | 3159a | 3600a | 15496a | 7677a |
| | JM585 | N0 | 1060a | 1725e | 3600a | 13941b | 7556a | 1060a | 1725e | 3600a | 10630c | 4245b |
| | | N120 | 1060a | 2297d | 3600a | 14963a | 8006a | 1060a | 2297d | 3600a | 11963b | 5006ab |
| | | N180 | 1060a | 2585c | 3600a | 15296a | 8051a | 1060a | 2585c | 3600a | 12611ab | 5366a |
| | | N240 | 1060a | 2873b | 3600a | 15430a | 7897a | 1060a | 2873b | 3600a | 12833a | 5300a |
| | | N300 | 1060a | 3159a | 3600a | 15354a | 7535a | 1060a | 3159a | 3600a | 12666ab | 4847ab |

Note: Different letters following data of same trait and variety indicate significant differences between nitrogen treatments ($P < 0.05$).

Gaocheng, Xinle and Zhaoxian. As for 2019–2020, net return obtained from KN199 was significantly the highest under the N180 treatment being higher by 3.49%–20.48%, 3.79%–77.32%, 1.84%–23.13% than other treatment at Gaocheng, Xinle and Zhaoxian. Net return obtained from JM585 was significantly the highest under the N180 treatment being higher by 4.33%–66.96%, 3.02%–68.38%, 1.25%–26.41% than other treatment at Gaocheng, Xinle and Zhaoxian.

## Optimization of variety and N application rate based on yield and economic return

The data collected in this study over two growing seasons were comprehensively analyzed by considering the effects of different fertilization rates. N application rate was treated as an

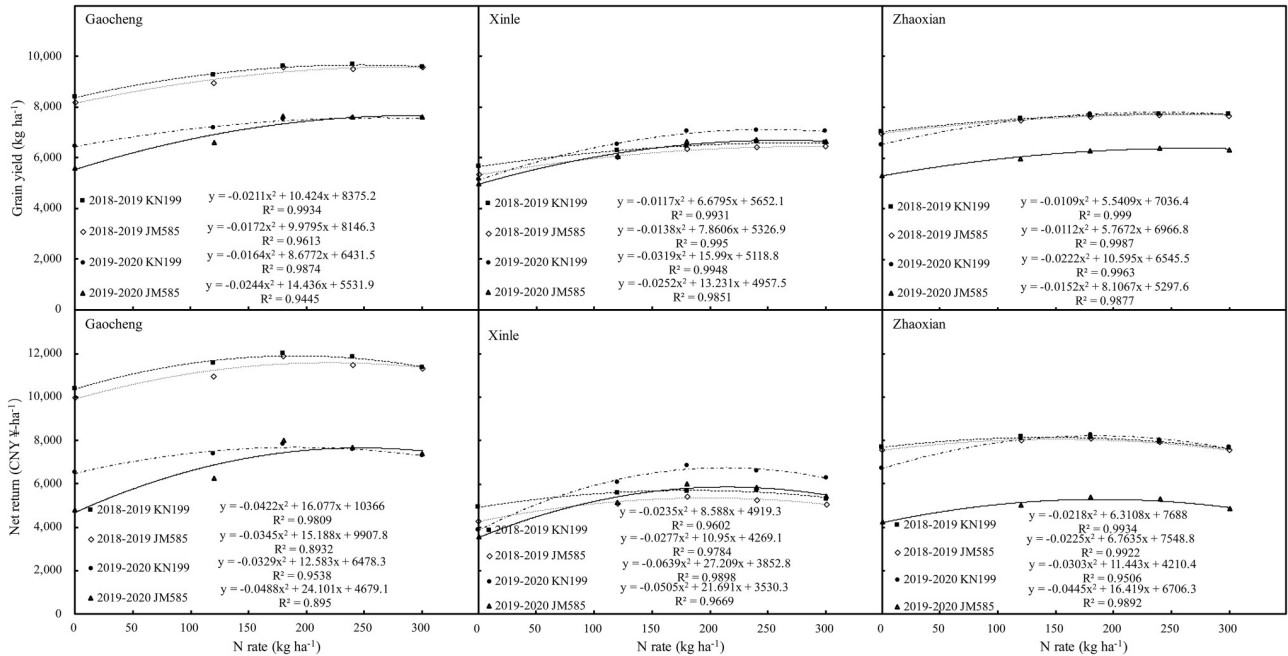

**Fig 6. Correlation of winter wheat grain yield and economic return to nitrogen fertilization rates.**

independent variable, and grain yield and net return were considered as response variables. The data were analyzed using the fertilizer effect function, and a unitary quadratic regression equation was determined, which was used to calculate the fertilizer amount required to maximize grain yield and net return (Fig 6). In 2018–2019, the maximum yield from KN199 was achieved with N application rates of 247.0, 285.5, and 254.2 N kg ha$^{-1}$ at Gaocheng, Xinle, and Zhaoxian, respectively. In turn, net return was maximized with the application of 190.5, 192.2, and 144.7 N kg ha$^{-1}$ at Gaocheng, Xinle and Zhaoxian, respectively. In the same period, the maximum yield from JM585 was achieved with N application rates of 290.1, 284.8, and 257.5 N kg ha$^{-1}$ at Gaocheng, Xinle, and Zhaoxian, respectively, while net return in this case was maximized with the application of 220.1, 246.9, and 150.3 N kg ha$^{-1}$ at Gaocheng, Xinle, and Zhaoxian, respectively. In turn, in the 2019–2020 growing season, the maximum yield from KN199 was achieved with N application rates of 264.6, 250.6, and 238.6 N kg ha$^{-1}$ at Gaocheng, Xinle, and Zhaoxian, respectively, while net return was maximized with the application of 191.2, 212.9, and 184.5 N kg ha$^{-1}$ at Gaocheng, Xinle, and Zhaoxian, respectively. Meanwhile, in the same period, the maximum yield from JM585 was achieved with N application rates of 295.8, 262.5, and 266.7 N kg ha$^{-1}$ at Gaocheng, Xinle, and Zhaoxian, respectively, while net return was maximized with the application of 223.3, 214.8, and 188.8 N kg ha$^{-1}$ at Gaocheng, Xinle, and Zhaoxian, respectively.

When the N application rate exceeded 180 kg ha$^{-1}$, grain yield did not increase significantly. Furthermore, at rates lower than the optimal N application rate, the net return of winter wheat increased with the application rate; however, when higher than the optimal rate, the return gradually decreased. Altogether, data indicated that the optimal N application rate in the Piedmont plain of the Taihang Mountains in China, ranged from 144.7 to212.9 kg ha$^{-1}$ for KN199 and from 150.3 to 247.0 kg ha$^{-1}$ for JM585.

## Discussion

### Effects of nitrogen application rate on LAI, aboveground biomass, grain yield, and harvest index of winter wheat

Variety, amount of irrigation, and fertilizer available for uptake directly affect the growth and development of wheat, thereby affecting final grain yield [26,27]. Suitable irrigation amount, N application rate, and variety can significantly increase LAI, aboveground biomass, and grain yield of winter wheat, while N uptake increases with increasing irrigation and N fertilization [28,29]. Previous studies found that LAI, biomass production, grain yield, and yield are positively affected by N fertilization, but negatively affected by water stress [30,31]. In the experiments reported herein, N application rate affected grain yield and LAI of the genotypes under evaluation; furthermore, genotypic responses varied with environment, i.e., with site and year [32]. LAI and aboveground biomass were significantly higher under the N180 treatment than under the N0'treatment across locations. When N input increased to N240, there were no significant differences in KN199, and when increased to N300, there were no significant differences in JM585 with respect to LAI or aboveground biomass. When the N application rate exceeded 180 kg ha$^{-1}$, grain yield did not increase significantly; as what increased mostly was the dry matter accumulated in vegetative organs, while HI decreased concomitantly. Guttieri et al. [33] studied the effect of N fertilization on the grain yield of different genotypes. Yue et al. [34] found that wheat grain yield increased with an increase in N supply but excess N did not increase grain yield nor grain N accumulation in five experimental sites. Consistently, Zhang et al. [35] showed that 190 kg N ha$^{-1}$ can be applied in Beijing, China, to maintain a steady yield for at least two years. In this study, grain yield increased with increasing N application rate but when it exceeded 180 kg ha$^{-1}$, the yield of the two varieties tested did not increase significantly. In 2019–2020, at the Gaocheng experimental location, the grain yield for KN199 and JM585 decreased by 21.27–23.18% and 20.02–26.69%, respectively, compared with the grain yield recorded in the 2018–2019 growing season. This was attributed to a reduction in the number of grains per panicle owing to freezing damage at the panicle differentiation stage and continuous rain during the flowering period (Fig 1).

### Effects of application rate on N accumulation and NUE

Excess N application results in reduced NUE and soil pollution [36]. Improving NUE facilitates the rational utilization of agricultural resources [37]. Zheng et al. [18] concluded that N accumulation in winter wheat increases with increasing N application rate; however, grain N accumulation decreases at N application rates above 240 kg N ha$^{-1}$. In this study, total N accumulation and grain N accumulation increased with increasing N application rate and when the latter exceeded 240 kg ha$^{-1}$, they remained increased in a non-significant way. Presumably because of differences in planting area and wheat variety. In addition, the results of this study showed that the N240 and N300 treatments increased total plant N uptake; however, grain N uptake and grain yield did not increase. Indeed, our results showed that maximum NHI was obtained under the N0'treatment in 2019–2020 at the three experimental sites; yet, conversely, the accompanying yields were the lowest. This may be due to frost damage during the panicle differentiation period and continuous rain during the flowering period of 2019–2020 (Fig 1), the number of grains per spike was reduced, which resulted in a decrease in grain yield. Several studies have reported a NUE below 30% for winter wheat [18,38]. Studies have shown that wheat varieties differ for NUE; high-N varieties (such as varieties with high uptake efficiency and utilization efficiency) also have high plant dry matter yields and grain yields under conditions of insufficient N supply [39]. In this study, NUE generally decreased and differed only

slightly with increasing N application rate across sites in both varieties; furthermore, it decreased in both with an increasing N application rate. NUE of KN199 and JM585 ranged within 30.4–39.89 and 28.11–41.99, and 29.2–42.29 and 27.22–37.78 in the 2018–2019 and 2019–2020 growing seasons, respectively. As it can be seen, NUE was higher in the first than in the second growing season, mainly owing to the high rainfall that caused the accumulation of dry matter in plant vegetative organs in the latter. Our results demonstrated that excessive N fertilization reduced grain N uptake, grain yield, and NUE. Therefore, environment-dependent site-specific genotype and N application rate recommendations should be promoted, in accordance to prevailing environmental conditions or seasonal expectations.

## Combined effects of variety and nitrogen application rate

Many researchers have elucidated the relationships among amount of irrigation, fertilizer input, and crop yield using a combination of multivariate regression and spatial analyses [40,41]. The optimal N application rate for wheat varies with soil conditions in the planting area. Previous studies on wheat have shown that an irrigation of 240 mm and an N application rate of 150–210 kg N ha$^{-1}$ can maintain higher productivity in the entire system with higher resource use efficiency and lower N$_2$O emissions in the NCP [11]. In this study, the relationships between fertilizer cost and crop yield were further analyzed and the net return for the maximum values of these parameters was assessed at three experimental sites. The study data suggest that the optimal N fertilizer rates to use at the Piedmont plain of the Taihang Mountains for the sustainable production of winter wheat varieties KN199 and JM585 are 144.7–212.9 and 150.3–247.0 kg ha$^{-1}$, respectively.

## Conclusions

Under limited irrigation conditions, LAI, dry matter accumulation, grain yield, and nitrogen accumulation of the two evaluated wheat varieties in the study sites increased with increasing N application rate; however, HI, NHI, and NUE decreased. The optimal fertilizer amounts for winter wheat varieties KN199 and JM585 varied within the ranges of 144.7–212.9 and 150.3–247.0 kg ha$^{-1}$, respectively, as these conferred the highest economic return, while facilitating water and fertilizer conservation, decreasing groundwater pollution risk, and maintaining high grain yield. The results of this study are of great significance for the scientific management of winter wheat fertilization in the Piedmont plains of the Taihang Mountains of China, which may be similar to many other arid and semi-arid winter wheat production areas around the world.

## Supporting information

**S1 Data.**
(XLSX)

## Acknowledgments

We highly appreciate associate professor Dong-xiao Li and other colleagues at Hebei Agricultural University who provided timely assist in this work.

## Author Contributions

**Data curation:** Pin Zhang.

**Funding acquisition:** Rui-qi Li.

**Investigation:** Pin Zhang, Yi-kang Qi.

**Methodology:** Pin Zhang, Hong-guang Wang.

**Project administration:** Hong-guang Wang.

**Supervision:** Rui-qi Li.

**Writing – original draft:** Pin Zhang.

**Writing – review & editing:** Pin Zhang, Jian-ning He, Wei-li Liang.

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
