## [Decision Letter · Decision Letter 0]

7 Sep 2021

PONE-D-21-25804Optimizing nitrogen fertilizer amount for best performance and highest economic return of winter wheat under limited water irrigation conditionsPLOS ONE

Dear Dr. Liang,

Thank you for submitting your manuscript to PLOS ONE. After careful consideration, we feel that it has merit but does not fully meet PLOS ONE’s publication criteria as it currently stands. Therefore, we invite you to submit a revised version of the manuscript that addresses the points raised during the review process.

We look forward to receiving your revised manuscript.

Kind regards,

Vassilis G. Aschonitis

Academic Editor

PLOS ONE

Journal Requirements:

4. Please upload a copy of Supporting Information Figures 1 to 6 and Tables 1 to 4. which you refer to in your text on page 32.

Reviewers' comments:

Reviewer's Responses to Questions

**Comments to the Author**

1. Is the manuscript technically sound, and do the data support the conclusions?

Reviewer #1: Yes

Reviewer #2: Yes

Reviewer #3: Yes

Reviewer #4: Partly

2. Has the statistical analysis been performed appropriately and rigorously? 

Reviewer #1: Yes

Reviewer #2: Yes

Reviewer #3: I Don't Know

Reviewer #4: Yes

3. Have the authors made all data underlying the findings in their manuscript fully available?

Reviewer #1: Yes

Reviewer #2: Yes

Reviewer #3: No

Reviewer #4: Yes

4. Is the manuscript presented in an intelligible fashion and written in standard English?

Reviewer #1: Yes

Reviewer #2: Yes

Reviewer #3: Yes

Reviewer #4: Yes

5. Review Comments to the Author

Reviewer #1: Recommendation: Major revision

The survey reports an interesting topic that points out the necessity of nitrogen fertilization optimization in the fields of winter wheat under limited irrigation and aiming a higher economic return. The manuscript is written in a good standard English style and a good statistical approach has been presented. I think the manuscript is of good quality and has a merit to be published in PLOS One when the recommended revision is successfully done. Following, a list of comments that will help authors improve their manuscript.

1. Page 2, line 28: Kindly remove “a” before few.

2. Page 3, line 40: Kindly add a space between “population” and “[1]”.

3. Page 3, lines 42–43: Kindly provide a reliable source (reference) for this statement.

4. Page 3, line 44: Kindly remove “mush too frequently” and adjust as follow: “farmers frequently use…”.

5. Page 4, line 58: Kindly adjust as follow: “is reported”.

6. Page 4, line 64: Kindly adjust as follow: “[16,17].

7. Page 5, line 67: Kindly define the abbreviation “NUE” at its first mention so that the reader can understand that you’re talking about the nitrogen use efficiency.

8. Page 5, line 68: Kindly add a space between “costs” and “[19]”.

9. Page 5, lines 74–78: I support the idea, but try to reformulate as the sentence is too long.

10. Page 6, lines 83–88: Kindly use the impersonal form for these sentences and avoid the first voice form.

11. Page 8, line 120: I’m okay with the arrangement; however, we can’t rely on unpublished works as a key reference.

12. Page 8, line 124: Kindly replace “In all” by “In sum”.

13. Page 9, lines 128, 130: Kindly adopt an homogeny term: either “growing season” or “cropping season”.

14. Page 9, line 131: Kindly add “respectively” after “11”.

15. Page 9, line 134: Kindly even use the atomic form of nitrogen, phosphorus, and potassium or their full name.

16. Page 9, line 143: Kindly put “Leaf area index (LAI)” as a subtitle in italic form.

17. Page 9, line 143: Kindly replace “in” by “from”.

18. Page 10, line 146: Same recommendation as point 13.

19. Page 10, line 152: Same recommendation as points 13 and 18.

20. Page 11, line 169–170: Kindly adjust as follow: “fertilizer cost”, and “other costs”.

21. Page 12, line 179: I suggest writing the full name of LAI here (leaf area index).

22. Page 12, line 180: You mean the single effect of location, variety and N rate not their combined effect (interaction); as the interaction of these three factors has no significant effect on the leaf area index in both growing seasons 2018–2019 and 2019–2020.

23. Page 12, line 183: Kindly add “in both growing seasons” after “N0 treatment”.

24. Page 12, lines 186–188: “However… treatments”: Kindly remove this section as discussing non significant results is not important.

25. Page 12, line 188: Kindly replace “However” by “Moreover”.

26. Page 12, line 189: Kindly adjust as follow: “in Xinle and Zhaoxian experiment locations”. At what growing season ?

27. Page 12, line 190: In comparison with which N treatment ? N0 ?

28. Page 12, line 191: Kindly remove “although… N300” for the same reason mentioned in point 24.

29. Page 13, Table 2: You may point out the interaction between location and variety in both seasons on leaf area index by naming it (L ˟ V) and the interaction between location and N rate in 2019–2020 season (L ˟ N).

30. Page 13, Table 2: Kindly define each abbreviation as a table footnote.

31. Page 13, line 195: Kindly remove “Location, variety, and N rate in 2018–2020 (Table 2).”

32. Page 13, line 195, Figure 3: Kindly correct the order of the locations graphs to be similar in the upper and the lower rows and capitalize the first letter of all locations names.

33. Page 13, line 196: In which growing season ?

34. Page 13, line 198: Kindly adjust as follow: “growing season”.

35. Page 13, line 201: Kindly adjust as follow: “N0’s one”.

36. Page 13, line 202: Kindly add that it was a significant increase in Xinle and Zhaoxian experimental locations.

37. Page 13, line 203: Same recommendation as in points 13, 18, and 19.

38. Page 14, line 204: Kindly mention if it was a significant increase in comparison with the control (N0 treatment), in which season, and by how much.

39. Page 14, line 205: Same recommendation as in point 13, 18, 19, and 37.

40. Page 14, line 206: Kindly adjust as follow: “in the previous season”.

41. Page 14, lines 208–210: Kindly mention the growing season.

42. Page 14, lines 213–214: Actually, you can mention in the previous sentences both seasons instead of splitting the idea into two.

43. Page 14, line 215: Kindly mention the effect of the interactions between location and variety, and location and N rate on the grain yield.

44. Page 14, lines 215–216: Actually, it was the case of all treatments in 2019–2020 season. It looks like it is correlated with environmental changes rather that fertilization effect.

45. Page 14, lines 217–218: Same recommendation as in points 13, 18, 19, 37, and 39.

46. Page 15, lines 219–227: Kindly mention the effect of the interactions between location and variety, and location and N rate in the growing season 2018–2019 on the harvest index.

47. Page 15, line 222: Kindly specify the N treatments.

48. Page 15, lines 223–225: Kindly mention the season and specify the N treatments.

49. Page 15, lines 225–226: Same recommendation as in point 42.

50. Page 15, line 229: Kindly mention the complete name of NUE in the title.

51. Page 15, lines 230–234: There are plenty of non-discussed results especially concerning the interactions between factors location, variety and N rate and their effect on the studied parameters.

52. Page 15, line 233: It is the case in both growing seasons.

53. Pages 15–16, lines 235–245: It is a reading of the results. Kindly try to compare the results rather than reading them.

54. Page 16, lines 246–248: Same recommendation as in points 42 and 49.

55. Pages 16–17, lines 251–252: Good discussion point, but kindly move it to the “Discussion” part.

56. Page 18, line 258: Kindly adjust as follow: “from the two tested winter wheat varieties”.

57. Page 18, line 261: Do you mean by “net return” the “economic return” ? Kindly adopt an homogeny term and mention when it increased and decreased.

58. Page 18, lines 262–270: Kindly follow the recommendation in the next point for this section.

59. Page 19, line 272, Table 4: Kindly make the Duncan test for this table to detect which treatments in both varieties and growing seasons give the most significant gross profit and economic return and lower costs. In the current way provided, we cannot rely on the results to judge.

60. Page 21, lines 299–302: The sentence is too heavy; kindly reformulate.

61. Page 21, lines 302–303: Kindly adjust as follow: “Altogether, data indicated that…”.

62. Page 22, line 311: Kindly adjust as follow: “[25,26]”.

63. Page 22, line 314: Kindly adjust as follow: “[27,28]”.

64. Page 22, line 316: Kindly adjust as follow: “[29,30]”.

65. Page 23, line 328: Add a space between “[34]” and “showed”.

66. Page 23, lines 334–336: Good discussion of the results !!

67. Page 24, lines 339–340: “Improving… resources”: Kindly provide a reliable source (reference) for this statement.

68. Page 24, line 352: Kindly adjust as follow: “[17,36]”.

69. Page 24, line 352: Kindly adjust as follow: “… NUE; high-N varieties…”

70. Page 24, line 353: Kindly mention the full name at the first mention of a parameter, and follow it by the abbreviation between brackets.

71. Page 25, line 359: Kindly adjust as follow: “As it can be seen…”

72. Page 25, lines 359–360: Same recommendation as in points 13, 18, 19, 37, 39, and 45.

73. Page 25, line 361: Kindly adjust as follow: “excessive” or “excess in”.

74. Page 25, line 362: Kindly remove “and” before “site-specific”.

75. Page 25, line 366: Kindly remove “The” from the paragraph title.

76. Page 25, line 369: Kindly adjust as follow: “[38,39]”.

77. Page 26, line 371: Is it an irrigation or a rainfall ? I guess you mean an annual or a seasonal rainfall. Kindly clarify.

78. Page 26, lines 373–378: Kindly use the impersonal form for these sentences.

79. Page 26, lines 381–385: The statement is right but is too long. Kindly reformulate.

80. Pages 26–27, lines 385–389: The statement is right but is too long. Kindly reformulate.

81. Page 27, lines 390–391: Kindly adjust as follow: “… management of winter wheat fertilization in the Piedmont plains…”

Reviewer #2: The topic is not novel; however, the research data covers the research objectives and the trails at various locations make it valuable.

My only concern is why the author didn't use the control (well-watered conditions) to compare the results under limited water conditions?

I will suggest to compare your results with the control (well-watered conditions) as well.

Reviewer #3: This study is not unique in itself as similar studies have been conducted in wheat as can been seen from many of the studies that authors have cited. However, this study does provide valuable and useful scientific information that can be helpful in moving toward sustainable wheat framing in the Piedmont plain region, hence I recommend the editor to consider accepting this manuscript for publication with revisions.

I do find the manuscript is well written and is technically sound but there are certain information that I found to be missing or not explained well enough for me to fully understand all the aspects of the material and method section, which I have listed below. Additionally, I was not been able to see any attached file with primary data (e.g gain yield data) used for various analysis which is required as per PLOS Data policy, please request authors to make those data available as supplemental file or deposited to a public repository, if not done so already.

Revision suggestion:

Material and Methods:

Experimental site description:

Line 115: Table 1 is showing NPK measurement prior to 2018 seeding, was similar measurement obtain for 2019 year before seeding? If yes, please add them

Experimental design

Line 118-124: Please reword and clarify the experimental design description. Was it complete randomized block design? Was the treatment randomized? How many subplots per main plot were there? If the main plot was 50 m. wide and sub-plots were 10m. wide with 1.0 m buffer between subplots then does that mean there were only 4 sub-plots per main plain (ie. Only 4 N treatments per main plot instead of 5 N treatment)

Line 122: What was the reason behind selecting those specific five N fertilizer rates?

Line 125: Was commercially available seed used both year for seeding?

Line 133: Please provide method of irrigation

Line 139: Please mention if topdressing of N was done after or before irrigation at the jointing stage.

Line 140: Provide information on which pesticides and herbicides were applied, mode of application and their dosage.

Measurements and calculations:

Line 153: Was grain moisture estimated after drying and was grain moisture used for adjusting grain yield?

Data analysis:

Line 173-174: Please provide the ANOVA model. Was year included in the model along with the location? If not, then does ANOVA result differ if year is added to the model.

Results:

Line 187-188: Was no significant differences in LAI for 2019-2020 observed due to higher residual N-P-K content in soil due to application? Was same exact field used in both years of the study?

Discussion:

Line 349-351: any specific environmental reason why grain per spike was reduced in 2019-2020?

Line 374-375: Will using available N in soil prior to seeding or fertilizer application improve calculation of required N fertilizer rate for each season more accurately and help improve economic return?

Reviewer #4: This study tried to find the appropriated nitrogen fertilizer on winter wheat with limited water resources. Authors measured the plant parameters and calculated the economic return. After reading this manuscript, I have several concerns. First, I am not sure the novelty of this manuscript. Second, authors should check the reference format/style carefully. Third, I felt that the cited reference in the main text is not the corrected one. Authors should check for it. In addition, authors used two wheat verities which KN199 has the high NUE and JM585 has the low NUE trait. However, in the table 3, there is no different between KN199 and JM585 on NUE. In addition, authors did not explain the table 2 very well which I think this table has lots of meaning. Besides, I have several comments as follows.

1. Authors should check the usage and grammar. For examples, line 44-45.

2. Line 46-49 and 64 and 159, I am not sure that cited references are relevant with the sentence.

3. Line 50, reference 78?

4. Line 70-71 and 74-78, what is the reference?

5. Line 79, what are the N absorption-related traits?

6. Line 88-90, I am not sure that this manuscript would provide the theoretical basis.

7. Line 119-121, authors used two different NUE wheat varieties. However, there is no reference on it. As I mentioned earlier, there is no different between KN199 and JM585 on NUE. Authors should explain for that.

8. Line 167, it should be E instead of Eb

9. In the result section, authors used the combined data in the text but used the separate data in the table. It is hard to judge the number.

10. Line 233, it should be 2018-2020.

11. Line 238, it should be 152.77-237.42 instead of 152.77-237.42.

12. Line 241, it should be 131.23-187.44 instead of 152.77-237.42.

13. Line 242-245, authors should check the number carefully.

14. Line 245, based on the statistical data, author cannot say that (there is no difference).

15. Line 250-252, there is no comparison between the data in 2018-2019 and 2019-2020. Thus, author cannot say the difference.

6. PLOS authors have the option to publish the peer review history of their article (what does this mean?). If published, this will include your full peer review and any attached files.

Reviewer #1: No

Reviewer #2: No

Reviewer #3: No

Reviewer #4: No

---

## [Author Response · Author response to Decision Letter 0]

29 Sep 2021

Dear Editor,

We appreciate the opportunity to revise the manuscript entitled " Optimizing nitrogen fertilizer amount for best performance and highest economic return of winter wheat under limited irrigation conditions" (Manuscript ID PONE-D-21-25804). We would like to express our sincere thanks to you and the reviewers for your constructive and valuable comments, which are very helpful for improving the manuscript. We have revised the text as described below and also provide point-by-point response to each comment. Changes in the revised manuscript are indicated in yellow font. We sincerely hope that this revised version of the manuscript is qualified for publication in PLOS ONE. 

Reviewer #1: Recommendation: Major revision 

The survey reports an interesting topic that points out the necessity of nitrogen fertilization optimization in the fields of winter wheat under limited irrigation and aiming a higher economic return. The manuscript is written in a good standard English style and a good statistical approach has been presented. I think the manuscript is of good quality and has a merit to be published in PLOS One when the recommended revision is successfully done. Following, a list of comments that will help authors improve their manuscript.

1. Page 2, line 28: Kindly remove “a” before few.

Response: Thank you for your valuable comment based on which we have revised the manuscript accordingly. We removed it at revised manuscript.

2. Page 3, line 40: Kindly add a space between “population” and “[1]”.

Response: Thank you for your valuable comment based on which we have revised the manuscript accordingly. We added it at revised manuscript.

3. Page 3, lines 42–43: Kindly provide a reliable source (reference) for this statement.

Response: Thank you for your valuable comment based on which we have revised the manuscript accordingly. We added reference [3].

[3] Chinese Statistical Bureau. China Statistical Yearbook. Beijing: China Statistics Press; 2020.

4. Page 3, line 44: Kindly remove “mush too frequently” and adjust as follow: “farmers frequently use…”.

Response: Thank you for your valuable comment based on which we have revised the manuscript accordingly. 

5. Page 4, line 58: Kindly adjust as follow: “is reported”.

Response: Thank you for your valuable comment based on which we have revised the manuscript accordingly.

6. Page 4, line 64: Kindly adjust as follow: “[16,17].

Response: Thank you for your valuable comment based on which we have revised the manuscript accordingly.

7. Page 5, line 67: Kindly define the abbreviation “NUE” at its first mention so that the reader can understand that you’re talking about the nitrogen use efficiency.

Response: Thank you for your valuable comment based on which we have revised the manuscript accordingly. We modified to nitrogen use efficiency (NUE).

8. Page 5, line 68: Kindly add a space between “costs” and “[19]”.

Response: Thank you for your valuable comment based on which we have revised the manuscript accordingly. We added it at revised manuscript.

9. Page 5, lines 74–78: I support the idea, but try to reformulate as the sentence is too long.

Response: Thank you for your valuable comment based on which we have revised the manuscript accordingly.

10. Page 6, lines 83–88: Kindly use the impersonal form for these sentences and avoid the first voice form.

Response: Thank you for your valuable comment based on which we have revised the manuscript accordingly.

11. Page 8, line 120: I’m okay with the arrangement; however, we can’t rely on unpublished works as a key reference.

Response: Thank you for your valuable comment. This work has been accepted by Journal of Triticeae Crops. We added this reference at the revised manuscript.

21. Zhang P, Wang HG, Fang et al. Q. Diference of Low-Nitrogen Tolerance in Sedling Quality and Activity of Nitrogen Metabolism Related Key Enzymes betwen Nitrogen Responding Types of Winter Wheat Varieties.2020; Journal of Triticeae Crops.41(9). doi:107.606/ji.sn1.009-1041

12. Page 8, line 124: Kindly replace “In all” by “In sum”.

Response: Thank you for your valuable comment based on which we have revised the manuscript accordingly.

13. Page 9, lines 128, 130: Kindly adopt an homogeny term: either “growing season” or “cropping season”.

Response: Thank you for your valuable comment based on which we have revised the manuscript accordingly. We changed "growth" to "growing".

14. Page 9, line 131: Kindly add “respectively” after “11”.

Response: Thank you for your valuable comment based on which we have revised the manuscript accordingly. We added “respectively” after “11”.

15. Page 9, line 134: Kindly even use the atomic form of nitrogen, phosphorus, and potassium or their full name.

Response: Thank you for your valuable comment based on which we have revised the manuscript accordingly. We changed "P and K" to " phosphorus and potassium "

16. Page 9, line 143: Kindly put “Leaf area index (LAI)” as a subtitle in italic form.

Response: Thank you for your valuable comment based on which we have revised the manuscript accordingly. We put “Leaf area index (LAI)” as a subtitle in italic form. We also made similar changes to the other measurements in the manuscript.

17. Page 9, line 143: Kindly replace “in” by “from”.

Response: Thank you for your valuable comment based on which we have revised the manuscript accordingly. We changed "in " to " from".

18. Page 10, line 146: Same recommendation as point 13.

Response: Thank you for your valuable comment based on which we have revised the manuscript accordingly. We changed "growth" to "growing". 

19. Page 10, line 152: Same recommendation as points 13 and 18.

Response: Thank you for your valuable comment based on which we have revised the manuscript accordingly. We changed "growth" to "growing".

20. Page 11, line 169–170: Kindly adjust as follow: “fertilizer cost”, and “other costs”.

Response: Thank you for your valuable comment based on which we have revised the manuscript accordingly. We changed "fertilizer input "to" fertilizer cost" and "other input "to" other cost".

21. Page 12, line 179: I suggest writing the full name of LAI here (leaf area index).

Response: Thank you for your valuable comment based on which we have revised the manuscript accordingly. We changed "LAI" to "Leaf area index".

22. Page 12, line 180: You mean the single effect of location, variety and N rate not their combined effect (interaction); as the interaction of these three factors has no significant effect on the leaf area index in both growing seasons 2018–2019 and 2019–2020.

Response: Thank you for your valuable comment. Yes, it is the single effect of location, variety and N rate on LAI.

23. Page 12, line 183: Kindly add “in both growing seasons” after “N0 treatment”.

Response: Thank you for your valuable comment based on which we have revised the manuscript accordingly. We added “in 2018-2019 growing seasons” after “N0 treatment”.

24. Page 12, lines 186–188: “However… treatments”: Kindly remove this section as discussing non significant results is not important.

Response: Thank you for your valuable comment based on which we have revised the manuscript accordingly. We removed this section.

25. Page 12, line 188: Kindly replace “However” by “Moreover”.

 Response: Thank you for your valuable comment based on which we have revised the manuscript accordingly. We replaced “However” by “Moreover”.

26. Page 12, line 189: Kindly adjust as follow: “in Xinle and Zhaoxian experiment locations”. At what growing season?

 Response: Thank you for your valuable comment based on which we have revised the manuscript accordingly. We added the growing season in 2019-2020. 

27. Page 12, line 190: In comparison with which N treatment? N0?

 Response: Thank you for your valuable comment based on which we have revised the manuscript accordingly. This is compared with N180 treatment and we added it in the manuscript.

28. Page 12, line 191: Kindly remove “although… N300” for the same reason mentioned in point 24.

 Response: Thank you for your valuable comment based on which we have revised the manuscript accordingly. We removed this section.

29. Page 13, Table 2: You may point out the interaction between location and variety in both seasons on leaf area index by naming it (L ˟ V) and the interaction between location and N rate in 2019–2020 season (L ˟ N). 

 Response: Thank you for your valuable comment based on which we have revised the manuscript accordingly. We added the L×V significantly affected the LAI in 2018–2020, and L×V only affected the LAI in 2019–2020.

30. Page 13, Table 2: Kindly define each abbreviation as a table footnote.

Response: Thank you for your valuable comment based on which we have revised the manuscript accordingly. We added the full names of the abbreviations in the table as footnotes.

31. Page 13, line 195: Kindly remove “Location, variety, and N rate in 2018–2020 (Table 2).”

Response: Thank you for your valuable comment. This is about Location, variety, and N rate significantly affected aboveground biomass in the two growing seasons considered herein. We changed the expression into “Location, variety, and N rate significantly affected aboveground biomass in the two growing seasons considered herein (Table 2).”

32. Page 13, line 195, Figure 3: Kindly correct the order of the locations graphs to be similar in the upper and the lower rows and capitalize the first letter of all locations names. 

Response: Thank you for your valuable comment based on which we have revised the manuscript accordingly. We corrected the upper and the lower rows and capitalized the first letter of all location names.

33. Page 13, line 196: In which growing season?

Response: Thank you for your valuable comment based on which we have revised the manuscript accordingly. We added the growing season in 2018-2019.

34. Page 13, line 198: Kindly adjust as follow: “growing season”.

Response: Thank you for your valuable comment based on which we have revised the manuscript accordingly. We changed it to “growing season”.

35. Page 13, line 201: Kindly adjust as follow: “N0’s one”.

Response: Thank you for your valuable comment based on which we have revised the manuscript accordingly. We changed it into “N0’ treatment”.

36. Page 13, line 202: Kindly add that it was a significant increase in Xinle and Zhaoxian experimental locations.

Response: Thank you for your valuable comment based on which we have revised the manuscript accordingly. We added the significant increase in Xinle and Zhaoxian experimental locations.

37. Page 13, line 203: Same recommendation as in points 13, 18, and 19.

Response: Thank you for your valuable comment based on which we have revised the manuscript accordingly. We changed "growth" to "growing".

38. Page 14, line 204: Kindly mention if it was a significant increase in comparison with the control (N0 treatment), in which season, and by how much.

Response: Thank you for your valuable comment based on which we have revised the manuscript accordingly. This sentence expresses that the nitrogen application rate is increased from N180 to N240 and N300, and the increase in dry matter is mainly due to the increase in vegetative organs.

39. Page 14, line 205: Same recommendation as in point 13, 18, 19, and 37.

Response: Thank you for your valuable comment based on which we have revised the manuscript accordingly. We changed it to “growing season”.

40. Page 14, line 206: Kindly adjust as follow: “in the previous season”.

Response: Thank you for your valuable comment based on which we have revised the manuscript accordingly. We changed to “in the previous season”.

41. Page 14, lines 208–210: Kindly mention the growing season.

Response: Thank you for your valuable comment based on which we have revised the manuscript accordingly. We added the growing season in 2018-2019.

42. Page 14, lines 213–214: Actually, you can mention in the previous sentences both seasons instead of splitting the idea into two.

Response: Thank you for your valuable comment that could make a better description. Actually, we did try to merge the two seasons but it became somewhat cumbersome. We will keep improving our English proficiency.

43. Page 14, line 215: Kindly mention the effect of the interactions between location and variety, and location and N rate on the grain yield.

Response: Thank you for your valuable comment based on which we have revised the manuscript accordingly. We added the effect of the interactions between location and variety, and location and N rate on grain yield.

44. Page 14, lines 215–216: Actually, it was the case of all treatments in 2019–2020 season. It looks like it is correlated with environmental changes rather that fertilization effect.

Response: Thank you for your valuable comment. Yes, it is correlated with environmental changes. We explained that in the discussion section.

45. Page 14, lines 217–218: Same recommendation as in points 13, 18, 19, 37, and 39.

Response: Thank you for your valuable comment based on which we have revised the manuscript accordingly. We changed to “growing season”.

46. Page 15, lines 219–227: Kindly mention the effect of the interactions between location and variety, and location and N rate in the growing season 2018–2019 on the harvest index.

Response: Thank you for your valuable comment based on which we have revised the manuscript accordingly. We added the effect of the interactions between location and variety, and location and N rate in the growing season 2018–2019 on the harvest index.

47. Page 15, line 222: Kindly specify the N treatments.

Response: Thank you for your valuable comment based on which we have revised the manuscript accordingly. We added the N treatment.

48. Page 15, lines 223–225: Kindly mention the season and specify the N treatments.

Response: Thank you for your valuable comment based on which we have revised the manuscript accordingly. We added the growing season in 2018-2019.

49. Page 15, lines 225–226: Same recommendation as in point 42.

Response: Thank you for your valuable comment that could make a better description. Actually, we did try to merge the two seasons but it became somewhat cumbersome. We will keep improving our English proficiency.

50. Page 15, line 229: Kindly mention the complete name of NUE in the title.

Response: Thank you for your valuable comment based on which we have revised the manuscript accordingly. We added the complete name of NUE at its first mention.

51. Page 15, lines 230–234: There are plenty of non-discussed results especially concerning the interactions between factors location, variety and N rate and their effect on the studied parameters. 

Response: Thank you for your valuable comment based on which we have revised the manuscript accordingly. We added the interactions between factors location, variety and N rate and their effect on the studied parameters in the manuscript.

52. Page 15, line 233: It is the case in both growing seasons.

 Response: Thank you for your valuable comment based on which we have revised the manuscript accordingly. We changed to “2018-2020”.

53. Pages 15–16, lines 235–245: It is a reading of the results. Kindly try to compare the results rather than reading them. 

Response: Thank you for your valuable comment based on which we have revised the manuscript accordingly. We re-analyzed the data in the manuscript.

54. Page 16, lines 246–248: Same recommendation as in points 42 and 49.

Response: Thank you for your valuable comment that could make a better description. Actually, we did try to merge the two seasons but it became somewhat cumbersome. We will keep improving our English proficiency. 

55. Pages 16–17, lines 251–252: Good discussion point, but kindly move it to the “Discussion” part.

Response: Thank you for your valuable comment based on which we have revised the manuscript accordingly. We moved it to the “Discussion” part in the manuscript.

56. Page 18, line 258: Kindly adjust as follow: “from the two tested winter wheat varieties”. 

Response: Thank you for your valuable comment based on which we have revised the manuscript accordingly. We adjusted as “from the two tested winter wheat varieties” in revised manuscript.

57. Page 18, line 261: Do you mean by “net return” the “economic return” ? Kindly adopt an homogeny term and mention when it increased and decreased.

Response: Thank you for your valuable comment based on which we have revised the manuscript accordingly. We changed “economic return” to “net return”.

58. Page 18, lines 262–270: Kindly follow the recommendation in the next point for this section.

Response: Thank you for your valuable comment based on which we have revised the manuscript accordingly. We added Duncan test in Table 4 and revised the corresponding results analysis.

59. Page 19, line 272, Table 4: Kindly make the Duncan test for this table to detect which treatments in both varieties and growing seasons give the most significant gross profit and economic return and lower costs. In the current way provided, we cannot rely on the results to judge.

Response: Thank you for your valuable comment based on which we have revised the manuscript accordingly. We added Duncan test in Table 4 and revised the corresponding results analysis.

60. Page 21, lines 299–302: The sentence is too heavy; kindly reformulate.

Response: Thank you for your valuable comment based on which we have revised the manuscript accordingly. We modified the sentence in the manuscript.

61. Page 21, lines 302–303: Kindly adjust as follow: “Altogether, data indicated that…”.

Response: Thank you for your valuable comment based on which we have revised the manuscript accordingly. We adjusted as “Altogether, data indicated that…”.

62. Page 22, line 311: Kindly adjust as follow: “[25,26]”.

 Response: Thank you for your valuable comment based on which we have revised the manuscript accordingly. We adjusted as “[26,27]” (after ref 3 was added).

63. Page 22, line 314: Kindly adjust as follow: “[27,28]”.

Response: Thank you for your valuable comment based on which we have revised the manuscript accordingly. We adjusted as “[28,29]”

64. Page 22, line 316: Kindly adjust as follow: “[29,30]”.

Response: Thank you for your valuable comment based on which we have revised the manuscript accordingly. We adjusted as “[30,31]”

65. Page 23, line 328: Add a space between “[34]” and “showed”.

Response: Thank you for your valuable comment based on which we have revised the manuscript accordingly. We added a space between “[34]” and “showed”.

66. Page 23, lines 334–336: Good discussion of the results !!

Response: Thank you!!!

67. Page 24, lines 339–340: “Improving… resources”: Kindly provide a reliable source (reference) for this statement.

Response: Thank you for your valuable comment based on which we have revised the manuscript accordingly. We added reference [37].

[37] Guo Y, Chen Y, Searchinger TD, Zhou M, Pan D, et al.Air quality, nitrogen use efficiency and food security in China are improved by cost-effective agricultural nitrogen management.2020;Nature Food.1(10):648-658.doi:10.1038/s43016-020-00162-z

68. Page 24, line 352: Kindly adjust as follow: “[17,36]”.

Response: Thank you for your valuable comment based on which we have revised the manuscript accordingly. We adjusted as “[18, 38]”

69. Page 24, line 352: Kindly adjust as follow: “… NUE; high-N varieties…”

Response: Thank you for your valuable comment based on which we have revised the manuscript accordingly. We adjusted as follow: “… NUE; high-N varieties…”.

70. Page 24, line 353: Kindly mention the full name at the first mention of a parameter, and follow it by the abbreviation between brackets.

Response: Thank you for your valuable comment based on which we have revised the manuscript accordingly. We added the “complete name of” in revised manuscript.

71. Page 25, line 359: Kindly adjust as follow: “As it can be seen…”

Response: Thank you for your valuable comment based on which we have revised the manuscript accordingly. We adjusted as “As it can be seen…”.

72. Page 25, lines 359–360: Same recommendation as in points 13, 18, 19, 37, 39, and 45.

Response: Thank you for your valuable comment based on which we have revised the manuscript accordingly. We changed to “growing season”.

73. Page 25, line 361: Kindly adjust as follow: “excessive” or “excess in”.

Response: Thank you for your valuable comment based on which we have revised the manuscript accordingly. We changed to “excessive”.

74. Page 25, line 362: Kindly remove “and” before “site-specific”.

Response: Thank you for your valuable comment based on which we have revised the manuscript accordingly. We removed “and” before “site-specific”.

75. Page 25, line 366: Kindly remove “The” from the paragraph title.

Response: Thank you for your valuable comment based on which we have revised the manuscript accordingly. We removed “The” from the paragraph title.

76. Page 25, line 369: Kindly adjust as follow: “[38,39]”.

Response: Thank you for your valuable comment based on which we have revised the manuscript accordingly. We adjusted as “[40,41]”

77. Page 26, line 371: Is it an irrigation or a rainfall ? I guess you mean an annual or a seasonal rainfall. Kindly clarify.

Response: Thank you for your valuable comment based on which we have revised the manuscript accordingly. We clarified it is irrigation. Xin [11] showed the yield was the highest when the irrigation rate was 240 mm. 

78. Page 26, lines 373–378: Kindly use the impersonal form for these sentences.

Response: Thank you for your valuable comment based on which we have revised the manuscript accordingly. We modified the impersonal form for these sentences. 

79. Page 26, lines 381–385: The statement is right but is too long. Kindly reformulate.

Response: Thank you for your valuable comment based on which we have revised the manuscript accordingly. We checked and modified the sentence.

80. Pages 26–27, lines 385–389: The statement is right but is too long. Kindly reformulate.

Response: Thank you for your valuable comment based on which we have revised the manuscript accordingly. We checked and modified the sentence.

81. Page 27, lines 390–391: Kindly adjust as follow: “… management of winter wheat fertilization in the Piedmont plains…”

Response: Thank you for your valuable comment based on which we have revised the manuscript accordingly. We adjusted as follow: “… management of winter wheat fertilization in the Piedmont plains…”

Reviewer #2: The topic is not novel; however, the research data covers the research objectives and the trails at various locations make it valuable.

My only concern is why the author didn't use the control (well-watered conditions) to compare the results under limited water conditions?

I will suggest to compare your results with the control (well-watered conditions) as well.

Response: Thank you for your valuable comment. There are three reasons why there is no comparison between sufficient water conditions and restricted water conditions. The first, Winter wheat is the top irrigated crop in the North China Plain. There is a long history of over-irrigation on wheat using groundwater to achieve maximum yield in this region. However, long-term overexploitation of groundwater for irrigation has resulted in a decline of groundwater in the North China Plain, which has negatively affected the sustainable development of regional agriculture [5]. Therefore, limited irrigation has become a governmental policy encouraging farmers to save water and corresponding techniques have been developed and extended. The second, there are plenty of previous studies with wheat under both well-irrigated and limited water conditions in the piedmont plain. Based on previous studies, we designed and conducted this study under limited water conditions [41]. Finally, the major objective of this study is to identify the optimal nitrogen application rate under limited water conditions in the piedmont plain of the Taihang Mountains. 

5. Han S, Tian F, Liu Y, Duan X.Socio-hydrological perspectives of the co-evolution of humans and groundwater in Cangzhou, North China Plain.2017;Hydrology and Earth System Sciences.21(7):3619-3633.doi:10.5194/hess-21-3619-2017 

41. Si Z, Zain M, Mehmood F, Wang G, Gao Y, et al.Effects of nitrogen application rate and irrigation regime on growth, yield, and water-nitrogen use efficiency of drip-irrigated winter wheat in the North China Plain.2020;Agricultural Water Management.231(doi:10.1016/j.agwat.2020.106002 

Reviewer #3: This study is not unique in itself as similar studies have been conducted in wheat as can been seen from many of the studies that authors have cited. However, this study does provide valuable and useful scientific information that can be helpful in moving toward sustainable wheat framing in the Piedmont plain region, hence I recommend the editor to consider accepting this manuscript for publication with revisions.

I do find the manuscript is well written and is technically sound but there are certain information that I found to be missing or not explained well enough for me to fully understand all the aspects of the material and method section, which I have listed below. Additionally, I was not been able to see any attached file with primary data (e.g grain yield data) used for various analysis which is required as per PLOS Data policy, please request authors to make those data available as supplemental file or deposited to a public repository, if not done so already. 

Material and Methods:

Experimental site description:

Line 115: Table 1 is showing NPK measurement prior to 2018 seeding, was similar measurement obtain for 2019 year before seeding? If yes, please add them

Response: Thank you for your valuable comment based on which we have revised the manuscript accordingly. We added this information in revised manuscript.

Experimental design

Line 118-124: Please reword and clarify the experimental design description. Was it complete randomized block design? Was the treatment randomized? How many subplots per main plot were there? If the main plot was 50 m. wide and sub-plots were 10m. wide with 1.0 m buffer between subplots then does that mean there were only 4 sub-plots per main plain (ie. Only 4 N treatments per main plot instead of 5 N treatment)

Response: Thank you for your valuable comments based on which we have revised the manuscript accordingly. The experiment adopted a two-factor split plot design at three experimental locations. The variety treatments served as the main plots and N application rate served as the subplots. We added this experimental design description at revised manuscript. 

Thank you for your valuable comment. The sub-plots were 10 m long×5 m wide, the 1.0 m buffer is included in a 10-meter long plot. We modified the design description of the experiment in the revised manuscript.

Line 122: What was the reason behind selecting those specific five N fertilizer rates?

Response: Thank you for your valuable comment. We chose these five nitrogen treatments for the following reasons: First, previous studies have found that the optimal nitrogen application rate is between 150-240 kg ha-1 in different sites. Second, in the piedmont plain, farmers usually apply 240-300 kg ha-1 nitrogen on wheat. Therefore, 300 kg ha-1was set as the maximum and 120 kg ha-1 the minimum rate in order to cover all the possibilities.

Line 125: Was commercially available seed used both years for seeding?

Response: Thank you for your valuable comment. Yes, the two varieties are widely adopted by wheat farmers in the Piedmont plain of the Taihang Mountains and are easily available in seed markets. 

Line 133: Please provide method of irrigation

Response: Thank you for your valuable comment based on which we have revised the manuscript accordingly. We connected a water meter at the water outlet of the pump, then connect the water pipe to the plots in order to strictly control water quantity at 60 mm. We added this at revised manuscript.

Line 139: Please mention if topdressing of N was done after or before irrigation at the jointing stage.

Response: Thank you for your valuable comment. It was done before irrigation. We added this at revised manuscript.

Line 140: Provide information on which pesticides and herbicides were applied, mode of application and their dosage.

Response: Thank you for your valuable comment based on which we have revised the manuscript accordingly. We added this information at revised manuscript. 

Measurements and calculations:

Line 153: Was grain moisture estimated after drying and was grain moisture used for adjusting grain yield?

 Response: Thank you for your valuable comment based on which we have revised the manuscript accordingly. Grains were air-dried and weighed after harvest, samples were taken and oven-dried for measuring moisture content. Yields were normalized at 13 % moisture content. In the revised manuscript we added this information.

Data analysis:

Line 173-174: Please provide the ANOVA model. Was year included in the model along with the location? If not, then does ANOVA result differ if year is added to the model.

Response: Thank you for your valuable comment. The location is included in the model, but not the year. Based on your suggestion, we added year to the ANOVA model in revised manuscript.

Results:

Line 187-188: Was no significant differences in LAI for 2019-2020 observed due to higher residual N-P-K content in soil due to application? Was same exact field used in both years of the study?

 Response: Thank you for your valuable comments. The same plots were used in both years of the study. During the 2019-2020 growing season, LAI at N180 was significantly higher than N0 treatment. Moreover, in 2019-2020, at the Xinle and Zhaoxian experimental location, when N input increased from N180 to N240, LAI of JM585 increased significantly by 4.34% and 11.95%, respectively. LAI in 2019-2020 was higher than 2018-2019, mainly due to significantly higher rainfall in 2019-2020 (Fig.1).

Discussion:

Line 349-351: any specific environmental reason why grain per spike was reduced in 2019-2020?

Response: Thank you for your valuable comment. This may be because freezing damage during the panicle differentiation period and continuous rain during the flowering period in the 2019–2020 season (Fig.1). This information is in the discussion section.

Line 374-375: Will using available N in soil prior to seeding or fertilizer application improve calculation of required N fertilizer rate for each season more accurately and help improve economic return?

Response: Thank you for your valuable comment. The quantity of nitrogen that a season of wheat crop needs either from soil and fertilizer to achieve satisfactory yield in this region is 180-300kg ha-1. Fertilizer N inputs can be estimated if we know the available nitrogen in the soil. This on-demand input reduces the possibility of nitrogen leaching. However, given the very small farm and field size, it would be too expensive to obtain such data for each field or even each farm. 

Reviewer #4: This study tried to find the appropriated nitrogen fertilizer on winter wheat with limited water resources. Authors measured the plant parameters and calculated the economic return. After reading this manuscript, I have several concerns. First, I am not sure the novelty of this manuscript. Second, authors should check the reference format/style carefully. Third, I felt that the cited reference in the main text is not the corrected one. Authors should check for it. In addition, authors used two wheat verities which KN199 has the high NUE and JM585 has the low NUE trait. However, in the table 3, there is no different between KN199 and JM585 on NUE. In addition, authors did not explain the table 2 very well which I think this table has lots of meaning. Besides, I have several comments as follows.

1. Authors should check the usage and grammar. For examples, line 44-45.

Response: Thank you for your valuable comment based on which we have revised the manuscript accordingly. We modified the grammar of sentences 44-45.

2. Line 46-49 and 64 and 159, I am not sure that cited references are relevant with the sentence.

Response: Thank you for your valuable comment based on which we have revised the manuscript accordingly. We rechecked the correctness of the references.

3. Line 50, reference 78?

Response: Thank you for your valuable comment. These are two references. I forgot to add ", "in the middle, which modified in the revised draft. 

4. Line 70-71 and 74-78, what is the reference?

Response: Thank you for your valuable comment. Line 69-71, It's actually the same reference. The language is not well organized. We have revised it in the revised manuscript. 

Mehrabi et al. [21] showed that there was no significant difference between 150 and 300 kg N ha-1 in Shiraz, Iran. The optimal N application rate is 150 kg N ha-1. 

Line 74-78, This sentence in the manuscript is a summary of the previous part and therefore does not includes references.

5. Line 79, what are the N absorption-related traits?

Response: Thank you for your valuable comment. The N absorption-related traits include grain yield, nitrogen accumulation in grains and plants, and nitrogen use efficiency of wheat. 

6. Line 88-90, I am not sure that this manuscript would provide the theoretical basis.

Response: Thank you for your valuable comment based on which we have revised the manuscript accordingly. We removed “theoretical basis”.

7. Line 119-121, authors used two different NUE wheat varieties. However, there is no reference on it. As I mentioned earlier, there is no different between KN199 and JM585 on NUE. Authors should explain for that.

Response: Thank you for your valuable comment. We added our published references to the revised manuscript. The following table is the results of the variety test in the paper.

First, a pot-culture variety test was carried out in artificial climate chamber under two nitrogen treatments of N0 and N240. In N0 treatment, only vermiculite (total nitrogen 0.9 g•kg-1) and tap water (total nitrogen 10.2 mg•L-1) provided nitrogen. Therefore, there are differences between artificial climate chamber environment and field environment. Second, the results of the preliminary test were classified into low nitrogen tolerance high yield type and nitrogen sensitive type according to low nitrogen tolerance index. Therefore, in the revised manuscript, ‘high NUE’ and ‘low NUE’ were modified as ‘low nitrogen tolerance and high yield’ and ‘nitrogen sensitivity’. Third, the selection of varieties should first consider low nitrogen tolerance index, wide adoption in the piedmont plain. Therefore, KN199 and JM585 are selected as experimental materials.

8. Line 167, it should be E instead of Eb

Response: Thank you for your valuable comment based on which we have revised the manuscript accordingly. 

9. In the result section, authors used the combined data in the text but used the separate data in the table. It is hard to judge the number.

Response: Thank you for your valuable comment based on which we have revised the manuscript accordingly. We modified it in the revised manuscript.

10. Line 233, it should be 2018-2020.

Response: Thank you for your valuable comment based on which we have revised the manuscript accordingly. We changed to 2018-2020.

11. Line 238, it should be 152.77-237.42 instead of 152.77-237.42.

Response: Thank you for your valuable comment based on which we have revised the manuscript accordingly. We corrected it in the revised manuscript.

12. Line 241, it should be 131.23-187.44 instead of 152.77-237.42.

Response: Thank you for your valuable comment based on which we have revised the manuscript accordingly. We corrected it in the revised manuscript.

13. Line 242-245, authors should check the number carefully.

Response: Thank you for your valuable comment based on which we have revised the manuscript accordingly. We made a careful check of the number in the manuscript.

14. Line 245, based on the statistical data, author cannot say that (there is no difference).

Response: Thank you for your valuable comment based on which we have revised the manuscript accordingly. We checked and modified it in the revised manuscript.

15. Line 250-252, there is no comparison between the data in 2018-2019 and 2019-2020. Thus, author cannot say the difference.

Response: Thank you for your valuable comment based on which we have revised the manuscript accordingly. We checked and modified it in the revised manuscript.

More detailed response with tables and figures can be found in the uploaded file "response to reviewers"

---

## [Decision Letter · Decision Letter 1]

18 Oct 2021

PONE-D-21-25804R1Optimizing nitrogen fertilizer amount for best performance and highest economic return of winter wheat under limited irrigation conditionsPLOS ONE

Dear Dr. Liang,

Thank you for submitting your manuscript to PLOS ONE. After careful consideration, we feel that it has merit but does not fully meet PLOS ONE’s publication criteria as it currently stands. Therefore, we invite you to submit a revised version of the manuscript that addresses the points raised during the review process.

We look forward to receiving your revised manuscript.

Kind regards,

Vassilis G. Aschonitis

Academic Editor

PLOS ONE

Journal Requirements:

Reviewers' comments:

Reviewer's Responses to Questions

**Comments to the Author**

1. If the authors have adequately addressed your comments raised in a previous round of review and you feel that this manuscript is now acceptable for publication, you may indicate that here to bypass the “Comments to the Author” section, enter your conflict of interest statement in the “Confidential to Editor” section, and submit your "Accept" recommendation.

Reviewer #1: (No Response)

Reviewer #2: All comments have been addressed

2. Is the manuscript technically sound, and do the data support the conclusions?

Reviewer #1: Yes

Reviewer #2: Yes

3. Has the statistical analysis been performed appropriately and rigorously? 

Reviewer #1: Yes

Reviewer #2: Yes

4. Have the authors made all data underlying the findings in their manuscript fully available?

Reviewer #1: Yes

Reviewer #2: Yes

5. Is the manuscript presented in an intelligible fashion and written in standard English?

Reviewer #1: Yes

Reviewer #2: Yes

6. Review Comments to the Author

Reviewer #1: Recommendation: Minor revision

The revised manuscript’s version represents an improved one compared to the original submission. Authors replied to all my queries in their point-by-point answers. I am glad to say that authors took into consideration the detailed recommendations proposed. Several clarifications of previously ambiguous points were presented. Major improvements in the Results part were noticed; however, minor modifications are required in this section. Overall, the manuscript shows a higher value in its current form. I think the manuscript has a merit to be published in Plos One when the recommended minor revision is successfully done. Following, some comments that will help authors improve more and more their manuscript.

1. Page 9, lines 134–136: Kindly adopt the following manner when stating the methods of fertilization and irrigation: “N fertilizer was spread… a water meter was connected… then the water pipe was connected to the plots…”

2. Page 10, line 144: Kindly adjust as follow: “pesticides”.

3. Page 13, lines 197–199: Kindly adjust the sentence as follow: “Moreover, in 2019-2020, at Xinle and Zhaoxian experimental locations, the LAI of JM585 increased significantly by 4.34% and 11.95%, respectively when the N input increased from N180 to N240”.

4. Page 15, lines 213–214: Kindly adjust the sentence as follow: “aboveground biomass significantly increased in Xinle and Zhaoxian experimental locations.”

5. Page 18, lines 259–260: Kindly adjust the sentence as follow: “total N accumulation significantly increased in the three experimental locations”.

6. Page 20–21, lines 284–292: Kindly compare statistically your results in this section. In other words, compare your results in the following manner: “Net return was significantly the highest (you state the growing season, location and variety) being higher by … (you state the range between the minimum net return and the maximal one). In this manner, you give your valuable study and its results their right and you avoid reading the table. I’m glad seeing the ANOVA test added for gross profit and economic return as recommended and your results look now more valuable. Good work !!

However, kindly add the ANOVA test also for water cost and cost cultivation (all “a” as being the same). Kindly add also the ANOVA test for fertilizer cost in order to detect if there is significance between all N fertilization rates or not, and discuss the obtained results.

7. Page 26, lines 367–368: Kindly adjust the sentence as follow: “and when the latter exceeded 240 kg ha-1, they remained increased in a non-significant way”.

8. Page 26, line 373: Kindly adjust the sentence as follow: “This may be due to frost damage…”

Reviewer #2: (No Response)

7. PLOS authors have the option to publish the peer review history of their article (what does this mean?). If published, this will include your full peer review and any attached files.

Reviewer #1: No

Reviewer #2: No

---

## [Author Response · Author response to Decision Letter 1]

8 Nov 2021

Reviewer #1

We very much appreciate your time and kind detailed comments on the manuscript. Revises have been done accordingly.

1. Page 9, lines 134–136: Kindly adopt the following manner when stating the methods of fertilization and irrigation: “N fertilizer was spread… a water meter was connected… then the water pipe was connected to the plots…”

Response: Revised as advised.

2. Page 10, line 144: Kindly adjust as follow: “pesticides”.

Response: Revised as advised.

3. Page 13, lines 197–199: Kindly adjust the sentence as follow: “Moreover, in 2019-2020, at Xinle and Zhaoxian experimental locations, the LAI of JM585 increased significantly by 4.34% and 11.95%, respectively when the N input increased from N180 to N240”.

Response: Revised as advised.

4. Page 15, lines 213–214: Kindly adjust the sentence as follow: “aboveground biomass significantly increased in Xinle and Zhaoxian experimental locations.”

Response: Revised as advised.

5. Page 18, lines 259–260: Kindly adjust the sentence as follow: “total N accumulation significantly increased in the three experimental locations”.

Response: Revised as advised.

6. Page 20–21, lines 284–292: Kindly compare statistically your results in this section. In other words, compare your results in the following manner: “Net return was significantly the highest (you state the growing season, location and variety) being higher by … (you state the range between the minimum net return and the maximal one). In this manner, you give your valuable study and its results their right and you avoid reading the table. I’m glad seeing the ANOVA test added for gross profit and economic return as recommended and your results look now more valuable. Good work !!

However, kindly add the ANOVA test also for water cost and cost cultivation (all “a” as being the same). Kindly add also the ANOVA test for fertilizer cost in order to detect if there is significance between all N fertilization rates or not, and discuss the obtained results.

Response: Revised as advised.

7. Page 26, lines 367–368: Kindly adjust the sentence as follow: “and when the latter exceeded 240 kg ha-1, they remained increased in a non-significant way”.

Response: Revised as advised.

8. Page 26, line 373: Kindly adjust the sentence as follow: “This may be due to frost damage…”

Response: Revised as advised.

In addition,

1. In economic analysis, “benefit” was replaced with “return” and the changes were marked in yellow font.

2. The manuscript format was modified to meet the requirements of PLOS ONE. We added figure captions appear directly after the paragraph in which they are first cited in the manuscript. 

3. Table and figure captions were revised.

---

## [Editor Report · Decision Letter 2]

9 Nov 2021

Optimizing nitrogen fertilizer amount for best performance and highest economic return of winter wheat under limited irrigation conditions

PONE-D-21-25804R2

Dear Dr. Liang,

We’re pleased to inform you that your manuscript has been judged scientifically suitable for publication and will be formally accepted for publication once it meets all outstanding technical requirements.

Kind regards,

Vassilis G. Aschonitis

Academic Editor

PLOS ONE
---

## [Editor Report · Acceptance letter]

16 Nov 2021

PONE-D-21-25804R2 

Optimizing nitrogen fertilizer amount for best performance and highest economic return of winter wheat under limited irrigation conditions 

Dear Dr. Liang:

I'm pleased to inform you that your manuscript has been deemed suitable for publication in PLOS ONE. Congratulations! Your manuscript is now with our production department. 

Kind regards, 

on behalf of

Dr. Vassilis G. Aschonitis 

Academic Editor

PLOS ONE